ecology

coronavirus disease, behavioural plasticity, urban ecology, detectability, citizen science

**Author for correspondence:**
Oscar Gordo
e-mail: ogvilloslada@gmail.com

# Rapid behavioural response of urban birds to COVID-19 lockdown

Oscar Gordo[1], Lluís Brotons[2,3,4], Sergi Herrando[1,3] and Gabriel Gargallo[1]

[1]Catalan Ornithological Institute, ES-08019 Barcelona, Spain
[2]InForest Joint Research Unit (CTFC-CREAF), ES-25280 Solsona, Spain
[3]Centre of Ecological Research and Forestry Applications, ES-08193 Bellaterra, Spain
[4]Consejo Superior de Investigaciones Científicas, ES-08193 Cerdanyola del Vallès, Spain

OG, 0000-0003-3766-0566; LB, 0000-0002-4826-4457; SH, 0000-0002-5153-7970; GG, 0000-0003-3527-1997

Biodiversity is threatened by the growth of urban areas. However, it is still poorly understood how animals can cope with and adapt to these rapid and dramatic transformations of natural environments. The COVID-19 pandemic provides us with a unique opportunity to unveil the mechanisms involved in this process. Lockdown measures imposed in most countries are causing an unprecedented reduction of human activities, giving us an experimental setting to assess the effects of our lifestyle on biodiversity. We studied the birds' response to the population lockdown by using more than 126 000 bird records collected by a citizen science project in northeastern Spain. We compared the occurrence and detectability of birds during the spring 2020 lockdown with baseline data from previous years in the same urban areas and dates. We found that birds did not increase their probability of occurrence in urban areas during the lockdown, refuting the hypothesis that nature has recovered its space in human-emptied urban areas. However, we found an increase in bird detectability, especially during early morning, suggesting a rapid change in the birds' daily routines in response to quieter and less crowded cities. Therefore, urban birds show high behavioural plasticity to rapidly adjust to novel environmental conditions, such as those imposed by the COVID-19.

## 1. Introduction

Since the first human settlements some millennia ago, the anthropogenic transformation of the natural environment to build towns and cities has been a hallmark of humanity. During the last century, urbanization has experienced exponential growth across the world and it is expected to continue as more people will move from rural to urban areas [1–4]. As a result, urbanization has become one of the most important drivers of global change and a major threat to biodiversity [2,4–6]. Novel human-created environments, such as urban areas, represent a formidable challenge for organisms because the magnitude and peace of the environmental alterations imposed by humans usually exceed their limits of tolerance, leading to population shrinkage and extinction [5,7]. Urban challenges include dealing with chemical [3], acoustic [8,9] and light pollution [10,11], human disturbance [5,12], new pathogens [13,14] and predators [15,16], and human infrastructures [15,17]. However, some species are able to overcome these challenges and thrive in urban environments [4,7,12,18,19]. Therefore, a key question in urban ecology is how species cope with urbanization. Countless studies have demonstrated that adapting to urban environments implies some kind of phenotypical differentiation from non-urban relatives [7–9,12,18]. Indeed, organisms are forced to adjust their physiology, behaviour and life histories to the novel conditions imposed by the city [5,7]. However, little is known about the adaptive mechanisms allowing the differences observed between urban and non-urban dwellers [6,7].

Observed adjustments are mostly consistent with phenotypically plastic responses [12], but individual sorting and microevolutionary changes by divergent selection could be playing a role [4,5,7,18,19]. It may be that our inability to disentangle these mechanisms comes from a deficit of experimental studies in urban ecology [9], in spite of the fact that human-transformed environments often provide ready-made experiments. The current spread of the novel coronavirus disease (COVID-19) and its consequences represents an excellent example, as we are involuntarily involved in a major unintended social experiment.

After the declaration of the COVID-19 pandemic in March 2020 by the World Health Organization, most countries have implemented social and health measures unprecedented in recent history. These measures, aimed at containing the virus spread [20–23], have focused on social distancing and population confinement, as well as the cease of non-essential productive and social activities. Overall, the measures have contributed to a global diminishing of human activities [24]. This abrupt and dramatic disruption of most human social and economic activities has already had quantifiable effects on urban environments by marked reductions in air pollution [25–27] and noise [28–30]. One of the most notable and generalized measure has been applying certain degree of population lockdown, which renders our city streets empty and virtually silent. This situation provides a once-in-a-lifetime opportunity to study urban wildlife responses to less crowded, noisy and polluted cities and gain unprecedented mechanistic insights into how human activities affect wildlife [24,31–33]. As a result of the human lockdown, unusual observations of animals in urban areas worldwide have flooded the media and social networks planting in the social imaginary the idea that 'nature is getting back its space' (*sensu* [34]). Although plausible, this idea is, in most cases, based on anecdotal records, sometimes false [34,35], without any quantitative scientific investigation supporting such claim [24,33].

In this work, we aimed to assess the behavioural responses of birds to the sudden and drastic changes occurring in urban environments resulting from the COVID-19 lockdown in a densely populated area of northeastern Spain (Catalonia). Following China [23] and Italy [21], Spain was the third country worldwide to impose a severe population lockdown to stop COVID-19 spread. The declaration of the national emergency in 14 March 2020 by the Spanish Government imposed the strictest lockdown measures in Europe. Since then, social restrictions were alleviated progressively until the end of June (electronic supplementary material, figure S1 and electronic supplementary material, table S1). As in other parts of the world, this big halt of human activities has had significant environmental effects, with reduced air contamination and noise in Spanish cities [26,27,36,37]. The severity of the lockdown measures imposed in Spain make this country especially suitable to study COVID-19 lockdown effects in urban fauna, as they enjoyed exceptionally quieter towns and cities during many weeks.

We compared bird records collected during the first four weeks of the lockdown in towns and cities of Catalonia with the available records for the same region and dates since 2015. These historical records were used as baseline data. Our broad scale approach (hundreds of study sites covering and area of 32 000 km$^2$) at the community level (we studied 16 different species) allowed us a robust testing of two key questions:

(1) Did urban birds become more common in response to human-empty cities? It can be predicted that decreased human presence and disturbance allowed animals to occupy spaces that used to be above their fear tolerance thresholds [5,34,35]. Therefore, we expected a higher occurrence in 2020 compared to the historical records for the same urban areas. This effect is likely to be stronger for shyer species (i.e. urban adapters), who are less tolerant to human disturbance [5,12,38].

(2) Were urban birds more detectable as a consequence of quieter cities? It can be predicted that decreased anthropogenic noise increased the effective distance of among-bird communications [5,8,9,12] and made birds more easily perceived by observers [39,40]. Moreover, as the masking effect of human acoustic contamination mostly disappeared, we expected an increase in singing activity, including potential shifts in its timing, to profit from the new urban soundscape [8,9,41,42]. Therefore, we expected a higher detectability of urban birds during the lockdown than in previous years, with possible changes in the daily patterns of detection.

## 2. Material and methods

### (a) Bird data

On 14 March 2020, the Spanish Government declared the national emergency due to COVID-19 outbreak and imposed severe social restrictions. These restrictions included mandatory and permanent confinement of the population, border closure, limitations in public transport, online education, working from home whenever possible, and closure of non-essential business and public services. One day later, we launched the project '#JoEmQuedoACasa' (I stay at home) within the citizen science online platform Ornitho (www.ornitho.cat). This platform aims to collect wildlife records in Catalonia from birdwatchers and naturalists to improve knowledge of biodiversity in this region. Ornitho has been running since 2009 and has gathered more than 6.5 million records to date. The project launched during the lockdown and aimed to collect information about wildlife responses to the new environmental conditions resulting from people's confinement. In addition to this valuable information, the project was important to keep engaged birdwatchers in this citizen science programme by encouraging them to continue complete checklists submission, even during a period of constrained outdoor activities [43–45]. A complete checklist is a checklist with all identified species during any survey.

Lockdown surveys were conducted between 15 March and 13 April 2020. During these four weeks, people were subjected to the most restrictive conditions of mobility and consequently this period showed the most drastic reduction of human activities (electronic supplementary material, figure S1 and table S1). Therefore, lockdown checklists were carried out only from homes (e.g. balconies, rooftops or yards). To determine the effect of lockdown on bird behaviour, we also gathered all complete checklists available in Ornitho recorded during the same dates between 2015 and 2019. Surveyed sites were classified as urban or non-urban according to the 2017 land use/land cover map of Catalonia [46]. All surveys during the lockdown were in urban environments, except a few observers living in the countryside, which were excluded from the analyses. Therefore, we obtained three groups of checklists: urban lockdown, historical urban and historical non-urban, which contained a total of 126 315 bird records. Historical urban data represented baseline data, while historical non-urban data were included as control data without human disturbances. We used 5 years of historical data together

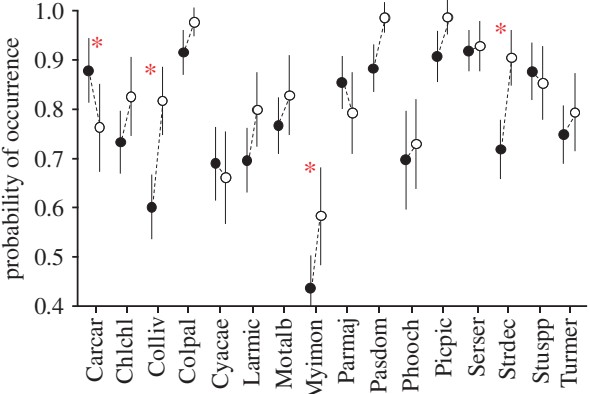

**Figure 1.** Probability of occurrence of birds in urban areas before (2015–2019, black dots) and during (2020; white dots) the COVID-19 lockdown. Asterisks indicate significant differences (*p*-value < 0.05). Error bars denote 95% confidence intervals. Acronyms for the species: Carcar *Carduelis carduelis*, Chlchl *Chloris chloris*, Colliv *Columba livia*, Colpal *Columba palumbus*, Cyacae *Cyanistes caeruleus*, Larmic *Larus michahellis*, Motalb *Motacilla alba*, Myimon *Myiopsitta monachus*, Parmaj *Parus major*, Pasdom *Passer domesticus*, Phooch *Phoenicurus ochruros*, Picpic *Pica pica*, Serser *Serinus serinus*, Strdec *Streptopelia decaocto*, Stuspp *Sturnus* spp., Turmer *Turdus merula*. (Online version in colour.)

to have a comparable number of checklists in urban areas to those recorded during the lockdown. By using several years of historical data we got a more representative baseline of the usual conditions previous to the COVID-19 pandemics, although we could not assess variability among years.

All checklists had associated basic information about the survey: site (geographical coordinates), date, hour, time invested (which was used as a proxy for sampling effort) and observer identity. We excluded checklists lasting greater than 3 h, as they might be discontinuous surveys. We also excluded those checklists started 1 h earlier or later than sunrise or sunset, respectively, as they represented nocturnal surveys. To correct for the adjustment of daylight saving time at the end of March, we rescaled recorded hours in civil time to the relevant daily sun events: sunrise, noon and sunset, which were established as −1, 0 and 1, respectively. Sunrise, noon and sunset were calculated for every geographical coordinate and date by the 'suncalc' library (v. 0.5.0) for R software [47]. Rescaling was calculated as the quotient between the difference of noon and checklist hour and the difference of sunrise or sunset and checklist hour, depending on whether the checklist started earlier or later than noon, respectively. This transformation allowed to fix the small bias caused by the longitudinal differences in sunrise and sunset across Catalonia as well as by the progressive day length increase during the study period. Not many observers recorded the number of individuals for each species. For this reason, we opted to work with presence/absence data.

We gathered 1289 complete bird checklists at 149 sites for the lockdown period. The number of replicated surveys per site and observer ranged from 1 to 91 (mean = 8.7, s.d. = 12.4). Historical records in urban areas were the scarcest: 1062 checklists in 410 sites with up to 48 replicates per site (mean = 2.6, s.d. = 5.2). As expected, data from non-urban areas were the most abundant, as observers usually preferred birdwatching in natural habitats. We gathered 5849 checklists from 3113 sites. Although one observer made 84 replicates for the same site, on average, observers in this group showed the lowest site fidelity (mean replicates = 1.9, s.d. = 3.6).

We selected data for the 16 most common sedentary urban species in Catalonia [48,49] (figure 1). We focused only on sedentary birds to avoid seasonal changes in occurrence and abundance associated with migration. Data from the common and the

spotless starlings (*Sturnus vulgaris* and *S. unicolor*, respectively) were merged as *Sturnus* spp. as both were not usually identified at the species level in most observations due to their high resemblance [50]. Both species are common, widespread, sympatric, and share similar habits and behaviour [49]. Thus, we did not expect important differences in their occurrence or detectability.

## (b) Statistical analyses

To disentangle the effects of individuals' presence (first question) and detection (second question) in our bird data, we used hierarchical occupancy models [51,52]. We considered as replicated surveys those checklists reported by the same observer within the same $1 \times 1$ km UTM cell. By combining observer and location, we avoided variability in detection rates due to observer expertise. We could assume confidently that observer experience was randomly distributed across our study area. The equations defining our model were:

$$\text{logit}(\Psi_j) = \beta_0 + \beta_1 \text{group}_{Lj} + \beta_2 \text{group}_{Uj} + \beta_3 \text{group}_{Nj}$$

and

$$\begin{aligned}\text{logit}(\rho_i) = {}& \alpha_0 + \alpha_1 \text{group}_{Li} + \alpha_2 \text{group}_{Ui} + \alpha_3 \text{group}_{Ni} + \alpha_4 \text{time}_i \\ & + \alpha_5 \text{time}_i \cdot \text{group}_{Li} + \alpha_6 \text{time}_i \cdot \text{group}_{Ui} + \alpha_7 \text{time}_i \cdot \text{group}_{Ni} \\ & + \alpha_8 f(\text{hour}_i) + \alpha_9 f(\text{hour}_i) \cdot \text{group}_{Li} \\ & + \alpha_{10} f(\text{hour}_i) \cdot \text{group}_{Ui} + \alpha_{11} f(\text{hour}_i) \cdot \text{group}_{Ni},\end{aligned}$$

where $\Psi_j$ is the occurrence of a species at site $j$ and $\rho_i$ is its detectability in the checklist $i$; groups $L$, $U$ and $N$ refer to lockdown, urban historical and non-urban, respectively; time refers to the duration of the survey; and hour refers to the starting hour. The hour was included as an unpenalized thin plate regression spline basis function ($f$) with five degrees of freedom because we expected that detectability could vary in a non-linear way along the day [53,54]. Interactions between group and time and between group and hour allowed us to model the effect of these two variables on detectability within each group. To test the significance of hour and interactions, we used log likelihood ratio tests. Basis functions were built by the smooth.construct function from package 'mgcv' (v. 1.8–22 [55]), while occupancy models were run with the occu function of package 'unmarked' (v. 0.12–3 [56]) for R.

## 3. Results

The probability of occurrence of a species during the lockdown did not differ significantly from the occurrence recorded in urban areas in previous years in 12 out of the 16 studied species after accounting for their imperfect detection (figure 1; electronic supplementary material, table S2). In the four species with significant differences, three increased their occurrence and one decreased it. As expected, most of the species (10) showed significant differences in their occurrence between lockdown and non-urban checklists (electronic supplementary material, table S2). On average, these species were approximately 15% more common in the lockdown checklists than in the non-urban checklists, confirming that most of the studied species were preferentially urban dwellers.

For most species (10), the probability of detection was higher in lockdown checklists than in historical urban ones, but this difference was not statistically significant in most cases (electronic supplementary material, figure S2 and table S3). Most species were less detectable in non-urban checklists than in urban ones.

As we predicted, detectability varied along the day in a nonlinear way for all species (figure 2; electronic

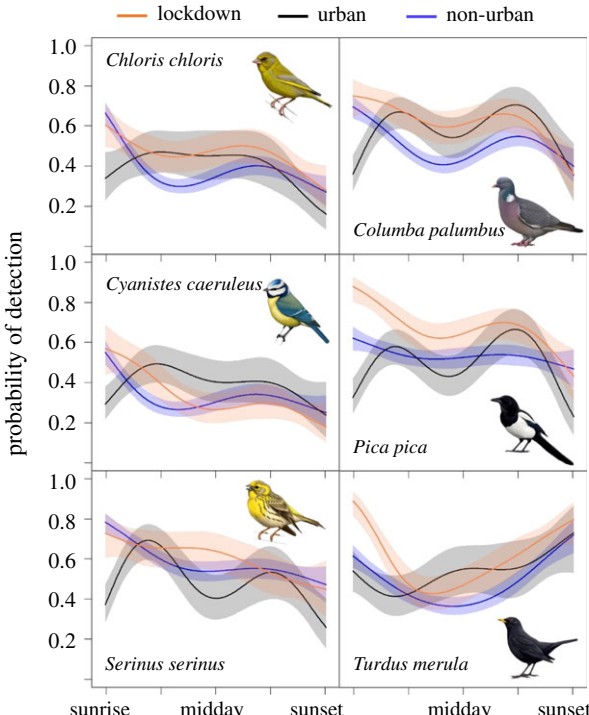

**Figure 2.** Variation in the probability of detection along the day for each group of data (collected during the lockdown, collected historically in urban sites and collected in non-urban environments). Shaded areas represent the 95% confidence intervals. See figure S2 in the electronic supplementary material for the rest of species. Bird illustrations by Martí Franch/Catalan Ornithological Institute. (Online version in colour.)

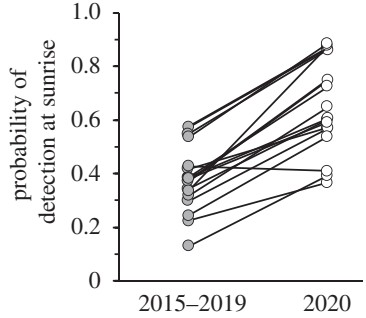

**Figure 3.** Probability of detection in urban environments at sunrise for the 16 studied bird species before (2015–2019) and during (2020) the COVID-19 lockdown.

supplementary material, figure S3). Except for two species, the pattern of daily variation in detectability was significantly different among groups (electronic supplementary material, table S4). A difference consistently found in most species was higher detectability in the first hours of the morning during the lockdown compared to the urban records from previous years (figures 2 and 3). In most species, during the lockdown detectability peaked at dawn and decreased until midday, while in the historical urban checklists the peak of detectability was around mid-morning. In fact, the pattern of detectability along the day in the lockdown group resembled more the non-urban pattern than the urban pattern in many species. Predicted detectability at sunrise by our models in the lockdown group was on average 27% higher than in the urban group (sign test: Z = 3.25, p-value = 0.001; figure 3).

As expected, in all but one species, chances of detection increased with longer surveys (electronic supplementary

material, figure S4 and table S5). In most of them (11), such a time effect was significantly different among groups (electronic supplementary material, table S4). Therefore, a certain increase of the sampling time implied a different increase in chances of detection in the lockdown, the historical urban and the historical non-urban groups for most species. For half of the species, the sampling time effect was significantly lower in the lockdown group than in the historical urban group (mean reduction of 17%; electronic supplementary material, table S5). This systematic reduction contrasts with the comparison of time effect between lockdown and non-urban groups, where for nine species there were significant differences between both groups, but such differences were disparate (mean change −0.8%; electronic supplementary material, table S5).

## 4. Discussion

Birds did not occur in higher rates in towns and cities during the lockdown than before it, contrary to the hypothesis that birds moved into the human emptied urban areas [33,34,38]. As the changes induced by the COVID-19 lockdown were drastic, sudden and of relatively short duration, they probably did not allow for colonization processes. The few species with a significant increase of their prevalence in urban surveys during lockdown were, interestingly, the ones that are mostly urban. As these species are not present in large numbers away from urban areas, they could hardly rely on non-urban source populations to occupy cities and towns during the lockdown. Their occurrence in a higher proportion of checklists during the lockdown could be due to the observers being constrained to survey from their homes. Urban checklists recorded during the lockdown probably were more focused on more extreme urban environments (i.e. core urban areas) than those checklists historically recorded in urban areas of Catalonia, which perhaps included a greater proportion of surveys in urban parks, landscaped plots or suburbs. These areas are considered urban areas from a land-use perspective, but they have more diversity of habitats at our working micro-scale (1 × 1 km), where the most urban exploiter species, such as feral pigeons (*Columba livia*), collared doves (*Streptopelia decaocto*) and monk parakeets (*Myiopsitta monachus*), may not find their most suitable niche. The absence of some non-essential activities during the lockdown, such as feral populations' management and culling [32], can be discarded as a cause for the increase of occurrence of these species. Despite being harmful invasive species [57,58], there is no management of the Catalan populations of Psittacidae yet. Culling of feral pigeons in big cities, such as Barcelona, was suspended due to ethical considerations in 2006 [59] and thus control measures for this species were unaffected by the lockdown. Finally, the collared dove colonized Catalonia naturally in the 1970s and occupied quickly most of the urban areas during the 1980s and 1990s [49,60]. Currently, its population grows at a 2% annual rate [61], which is too small to justify an 18% increase of population occupancy in 2020.

Birds changed their detectability pattern throughout the day as a consequence of the lockdown. In general, there was an increase in detection probability, which was especially marked in the early morning. As observed in non-urban habitats, detectability during the lockdown decreased from dawn onwards, while at the same urban locations detectability was

historically low at dawn and increased until reaching a peak 2 or 3 h later. It is interesting to note that the Eurasian blackbird, a model species in urban ecology studies [7,8,12,62,63], was the only exception to this pattern. Overall, many species showed a pattern of detection during the lockdown in urban areas more similar to that commonly seen in natural environments. Although in urban areas the 2020 detectability pattern was patently different from the 2015–2019 baseline level, such differences could have been partially enhanced because of comparing a single year against several pooled years. Nevertheless, it would be necessary to know whether or not there is significant among-years variability in bird daily routines. If that is the case, we would expect that behavioural responses to extreme events, such as the COVID-19 lockdown, would fall out of the normal variability. Unfortunately, we could not properly explore the between-years variation in detectability patterns due to sample size constraints.

Urban birds during lockdown may have shown this detectability peak at dawn, typical of non-urban habitats, because of a rapid behavioural response to adjust to the new environmental conditions imposed by the COVID-19 measures [64–68]. Birds rely heavily on acoustic communication. During reproduction, males sing to attract females and defend their territories, becoming highly conspicuous and detectable. COVID-19 lockdown was imposed just at the beginning of the breeding season, when singing activity was expected to be especially high [69]. Therefore, there was a strong pressure to time singing activity to the optimal moment of the day. This moment is dawn because the physical properties of the atmosphere enhance acoustic transmission [70,71] and consequently birds can reach the maximum audience. Thus, urban birds during the lockdown may have advanced their main period of singing activity to dawn, increasing their detection at those hours, similar to what is observed in non-urban areas.

During the lockdown, human presence and activities decreased drastically (electronic supplementary material, figure S1 and table S1), this being especially notable during rush hours, which almost disappeared [27,30,36]. During the spring in Spain, morning rush hour matches with the first hours of light, when birds are expected to be especially communicative [42,63,72]. The dramatic decrease in noise during the lockdown released early morning acoustic space that could be recovered by the dawn chorus. Empirical and experimental evidence demonstrates that urban birds avoid the masking effect of anthropogenic noise [8,9,41,42,73]. Our findings match these previous studies, but instead of advancing the dawn chorus [42,62,63,74], our historical urban data suggest that birds would delay their peak of activity (and consequently of detectability) to mid-morning. In our study context, this can be explained because civil and solar time are heavily decoupled in Spain since the country is located in the westernmost part of its time zone [75]. For this reason, if birds in Catalonia advance their activities before sunrise, they would be still suffering an important overlap with morning noisy human activities, such as commuting, school attendance, shop opening, etc. [63,72]. Hence, the best option for birds would be to delay the peak of activity to after the morning rush hour [73]. Moreover, most of the previous studies have been carried out in more northern latitudes [8,62,76], where climate conditions can still be severe at night in early spring. Under these circumstances, individual survival can be challenged by a strong nocturnal energy demand [77,78]. There, dawn singing can become a relevant and honest signal of the phenotypic quality of males, as

only those individuals in the best physical condition can undergo dawn fasting [62]. In Mediterranean regions, where spring nights are mild, the role of dawn singing as a signal of male quality might be less important. Attracting mates would be the prime objective for singing, and consequently males would be more pressed to place this activity when the interference of anthropogenic noise is at its lowest. Since sunrise, these lowest levels of noise are just after the morning rush hour (i.e. later than 9.00), when the physical properties of the air still keep sound attenuation and fluctuation low [70].

If birds have changed their behaviour, this adaptive, flexible behavioural response must have been mediated by phenotypic plasticity. Human lockdown was sudden and the environmental scenario in urban areas changed radically from one day to the next (electronic supplementary material, figure S1) [27,36,37]. This unprecedented social experiment imposed by COVID-19 allowed us to test and support the hypothesis of the high plasticity displayed by individuals living in urban areas in order to cope with a constantly changing environment [5–7,64]. However, this fast adaptive response might have been facilitated by a previous conditioning of birds to weekly rhythms of human activities. Birds change their behaviour from working to weekend days to match with human behaviours [41,73,79]. Therefore, birds could assimilate the lockdown as a very long and especially peaceful weekend, and consequently we may speculate that behavioural adjustments to novel lockdown conditions happened quickly (necessarily in less than one month). Nevertheless, it would be interesting to explore the long-lasting consequence at a community level of this environmental change [76]. Weekends are just 2 days long, while strict human lockdown lasted for at least 2 months in most regions of Spain. One may speculate that bolder and faster-adapting species are able to modify their behaviour on a weekly basis. However, during the lockdown, all species had enough time to habituate to the long-lasting new conditions. In fact, as we have demonstrated, all of them modified their daily patterns of detectability. Maybe the most urban species have benefited the least from this lockdown as their boldness and higher human tolerance was no longer an advantage in empty cities.

In addition to the birds' rapid behavioural response to the anomalous environmental conditions during the lockdown, observers had certainly enhanced opportunities to detect birds during this period. Urban areas were quieter than usual [27–30], improving the chances of listening the birds [34,39,40,70]. Moreover, the absence of people outdoors allowed for the display of shy and distrustful behaviours [5], facilitating bird observations, especially for those less singing species, such as the magpie (*Pica pica*) or the yellow-legged gull (*Larus michahellis*). Hence, birds' detectability during 2020 could be higher just as a by-product of a reduced interference in urban birdwatching of the human activities during the lockdown (e.g. traffic, pedestrians, factories, etc. [30,36,37]). However, these improved conditions for urban birdwatching were heavily constrained by the fact that observers were forced to stay at their homes and their sampling area was reduced to what they could see from there. Therefore, improved detection was to some extent counterbalanced by the limited scope from the survey sites. The observed effect in increased sampling time would support this hypothesis, as we demonstrated that the discovery rate in most species was slower during the lockdown than in the historical urban surveys.

The differences observed between urban and non-urban environments were expected as habitat configuration and bird densities are patently different between them. In fact, populations of urban exploiter birds show usually higher densities in cities than in rural or natural close areas [5–7], facilitating their detection in urban areas. Such differences may have serious consequences for monitoring schemes aiming to quantify wildlife occurrence and abundance by standardized protocols, as the assumption of equal detectability under similar circumstances is usually violated [39,52,53,80]. For instance, one sampling hour at dawn is not equivalent in terms of chances to detect a species in urban and non-urban habitats. Traditional protocols assume that the best moment to detect birds is early morning [70,81], which is actually true, but apparently only in natural conditions without human disturbance, as we have demonstrated here (figure 2). If the detectability peak in most urban populations is reached at mid-morning, their abundance would be systematically underestimated by usual sampling protocols based on early morning bird surveys. It does not matter whether this lower detectability in urban areas in early morning is caused by different daily routines of urban versus non-urban populations or by the masking effect of human activities in the city (or a combination of both). As there is an increased awareness about the importance of urban populations for bird conservation [7,48], it is necessary to ensure its accurate quantification, which may imply a redefinition of the most popular current census techniques [7]. Additionally, in this work, we demonstrated the utility of occupancy models and the necessity to account for imperfect detection [53,54].

The COVID-19 shutdown has revealed the stress, noise and pollution present in urban areas [25,26,28,29]. Under typical urban conditions, bird behaviour is apparently altered by our lifestyle, and thus the possibility to enjoy the natural values of our cities is notably diminished [7]. Our society should reflect on our urban lifestyle and how it affects the welfare of urban fauna and jeopardizes its conservation. As the world is becoming more urbanized and animals will be forced to live more often in anthropogenic environments [5–7], one way to ensure their adaptation as urban dwellers would be by reducing our noisier and more disturbing activities. Most importantly, not only urban populations of non-human animals would benefit, but also ourselves from quieter, more peaceful and less polluted cities.

Data accessibility. Data available from the Dryad Digital Repository: https://doi.org/10.5061/dryad.8w9ghx3kc [82].

Authors' contributions. G.G. had the original idea and designed the monitoring programme. O.G. and G.G. formulated formal hypotheses, and collected and arranged the necessary data for the study. O.G. analysed data. O.G. wrote the manuscript, with contributions from all the authors.

Competing interests. The authors declare no competing interests.

Funding. This study did not receive any funding.

Acknowledgements. We warmly thanks to the thousands of observers who shared their observations by Ornitho website, for their long-term involvement in this project and the special efforts undertaken during the lockdown. Ornitho project is supported by the Generalitat de Catalunya. Margarida Barceló-Serra provided valuable English edits and suggestions. We would also like to thank the anonymous reviewers and the editor for their comments.

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
