## [Peer Review File · Proceedings of the Royal Society B: Biological Sciences]

Review History

RSPB-2020-2513.R0 (Original submission)

Review form: Reviewer 1

Recommendation

Accept with minor revision (please list in comments)

Scientific importance: Is the manuscript an original and important contribution to its field?

Good

General interest: Is the paper of sufficient general interest?

Excellent

Quality of the paper: Is the overall quality of the paper suitable?

Excellent

Is the length of the paper justified?

Yes

Should the paper be seen by a specialist statistical reviewer?

No

Do you have any concerns about statistical analyses in this paper? If so, please specify them explicitly in your report.

No

It is a condition of publication that authors make their supporting data, code and materials available - either as supplementary material or hosted in an external repository. Please rate, if applicable, the supporting data on the following criteria.

Is it accessible?

Yes

Is it clear?

Yes

Is it adequate?

Yes

Do you have any ethical concerns with this paper?

No

Comments to the Author

The paper tests the effects of COVID-19 lockdown on bird activity in NE Spain. This is interesting because it takes profit of an unrepeatable “natural experiment” to test popular ideas on the effect of the lockdown on nature. The data is very robust, since it includes 16 different species over a very large area including many cities and towns. The paper is also interesting because allows to better understand patterns of bird daily activity in urban environments.

The paper is also original in using occupancy models with variable “observer” as if it was a “square”, allowing to quantify detectability. However, I wonder to which point the higher human activity in the first noisy hours of the day in “normal” urban days may reduce detectability and hence to provide a picture of lower activity at these hours, compared with more peaceful early hours in locked urban habitats. I assume that number of observers in 2020 was higher than in the unlocked years, so that detectability within the same observer could not be compared among years. Hence, some additional discussion or explanations would be appreciated on this point.

Given that lockdown was so sudden, could it be possible to check how many days could be necessary for birds to adapt their singing behaviour to the new situation?

Authors (lines 280-281) say that birds could assimilate the lockdown as a very long and especially peaceful weekend. Does it mean that they could change their singing behaviour one day to another? Some data and discussion on this would be interesting.

I have also a few minor comments:

Lines 48-49: In order to support your statement “However, some species are able to overcome these challenges and thrive in urban environments [4,8,13]”, you cite papers:

4. McDonnell MJ, Hahs AK. 2015 Adaptation and adaptedness of organisms to urban environments.

351 *Annu. Rev. Ecol. Evol. S.* 46, 261–280.

8. Slabbekoorn H, Ripmeester EA. 2007 Birdsong and anthropogenic noise: implications and applications for conservation. *Mol. Ecol.* 17, 72–83

13. Bradley CA, Altizer S. 2007 Urbanization and the ecology of wildlife diseases. *Trends Ecol. Evol.* 22, 95–102

Although paper 4 is right, since it is general and reviews on the topic, the other two are quite specific. I think that instead of these two I would advice you to cite: (Johnson and Munshi-South 2017; Szulkin et al. 2020).

In lines 73-76, authors cite paper 32 to back up the idea of some that nature is getting back its space, but I think they wanted to cite paper 35. This happens again in line 224. Please check.

In figure 2, authors display three lines; one is red and another one green. I would like to point out that about 8% of men are colour blinded, which means that cannot distinguish between these two lines. Following advice from many different sources (eg. (Allred et al. 2014) I would encourage authors to change the colours of their figure.

References

Allred SC, Schreiner WJ, Smithies O (2014) Colour blindness: still too many red-green figures. *Nature* 510:340. <https://doi.org/10.1038/510340e>

Johnson MTJ, Munshi-South J (2017) Evolution of life in urban environments. *Science* 358:eaam8327. <https://doi.org/10.1126/science.aam8327>

Szulkin M, Munshi-South J, Charmantier A (eds) (2020) *Urban evolutionary biology*: Na. Munshi-South. Oxford University Press, New York

Review form: Reviewer 2

Recommendation

Major revision is needed (please make suggestions in comments)

Scientific importance: Is the manuscript an original and important contribution to its field?

Good

General interest: Is the paper of sufficient general interest?

Good

Quality of the paper: Is the overall quality of the paper suitable?

Acceptable

Is the length of the paper justified?

Yes

Should the paper be seen by a specialist statistical reviewer?

No

Do you have any concerns about statistical analyses in this paper? If so, please specify them explicitly in your report.

Yes

It is a condition of publication that authors make their supporting data, code and materials available - either as supplementary material or hosted in an external repository. Please rate, if applicable, the supporting data on the following criteria.

Is it accessible?

Yes

Is it clear?

Yes

Is it adequate?

Yes

Do you have any ethical concerns with this paper?

No

Comments to the Author

This is a clever, straightforward study that uses data from a citizen science program to investigate whether reduced human activity in the context of a lockdown due to Covid-19 in a region of NE Spain led to shifts in bird activity patterns. The authors also investigate if detectability of some species changed throughout the day presumably due to associated rapid environmental changes such as noise or traffic reduction. Their main finding is that detectability of several bird species in urban settings at dawn increased by an average 27%, a pattern that may be seen as a convergence with activity patterns normally detected in non-urban environments.

This is a timely paper because it addresses a question that I assume numerous naturalists worldwide have been asking themselves. However, I have several questions that I would like to see clarified before I can recommend the paper for publication.

Gral comments:

My main concern is about the statistical consequences of comparing data from a single year (2020) to a pooled dataset of four previous years (2016-2019). The question is whether results would hold or, instead, whether among-year variation would erode significance in statistical results. I understand the authors possibly pooled together data from 2016-19 in order to have a sample size that was comparable to that of 2020 (which has altogether more bird checklists but a much lower number of sites) but it is possible such integration increased the chances of detecting a spurious statistical difference that would not be detected in comparisons year-by-year. It is important that the authors at least acknowledge the possible implications of this.

I wonder if multiple testing should be applied given that the authors are testing one independent hypothesis for each species. In the absence of multiple testing and taking a level of significance of $p=0.05$ when analyzing 16 species independently one would expect to detect, by chance, significant differences for at least one of these species.

Apart from the case of *C. carduelis*, an increase in the probability of occurrence was found for species that may be culled by local administrations like feral *C. livia* or the invasive *M. monachus*. Could the authors discuss if the lack of culling activities during lockdown could explain part of this variation? Finally, given recent range expansion of *S. decaocto* throughout Europe, one wonders whether there is some general trend of increased detection of this species throughout the years that could explain an increase in sightings when comparing 2020 to the previous four years?

Figure 1: As stated above, a most valuable comparison – and the plot I would really like to see – is a figure showing probability of occurrence and confidence intervals for every year (i.e. 2016 through 2020) that showed evidence for an increase in 2020. This would be the most fair comparison instead of using the average for the previous four years. Indeed, the same comment applies to Figure 3.

Specific comments:

-Author contributions:

The way this is currently written, it seems like two of the four authors have not contributed to any relevant stage in this research. Please double check this.

-Abstract:

In general, it appears to me there is an incorrect use of 'adaptation' throughout the ms. The paper would do equally fine using alternative wording and the authors will prevent themselves from

criticism by evolutionary ecologists. For example:

Lines 18-19: I would suggest to revisit this sentence given that ‘mechanisms of adaptation’ are not what is reported in this study which deals with changes in occurrence and daily patterns of activity.

Line 29: Same comment as above. Please consider editing the sentence “high behavioural plasticity to rapidly adapt to novel environmental conditions...”

-Introduction:

Line 53: Rephrase: “However, the mechanisms underlying the differences between urban and non-urban dwellers remain largely unknown”

Line 71: rephrase, less active cities?

Line 73: media and internet are not necessarily different things: e.g. social media commonly uses the network

Line 77: why is it relevant these are unexpected? By whom? The Intro could do as well with less subjective writing.

Line 87 and throughout the ms: please double check that ‘quitter’ is written instead of ‘quieter’.

Line 102: “Moreover, as the masking effect of human acoustic contamination mostly disappeared, we expected an increase in singing activity...”. I wonder if it would also be a valid alternative hypothesis to predict the opposite: if they communicate so well they might need less time to do the job.

Line 122: “to continue complete checklists (checklist with all identified species)”. It might be useful to provide here in the Intro – apart from the Methods section – a very brief specification of what a complete checklist is?

Line 127-128: The authors state all observations were made from private homes. Is this the same as saying all these observations corresponded to urban habitats? In case there are people providing data who live in – or next to – natural areas, should there be an additional category of observations classified as ‘lockdown non-urban’?

-Results:

Line 197: What is this “however” referring to?

Line 200: Excepting? Please reword.

Line 206-207: “In fact, the pattern of detectability along the day in the lockdown group resembled more to the non-urban pattern than to the urban pattern in many species” □ is there a formal way to provide a statistical test for this?

Line 211-214: I had a hard time understanding this sentence: what do “groups” refer to here? This is not properly indicated in Table S4 either.

Line 214: Please specify here what “time effect” refers to here.

-Discussion:

Line 223: please replace non-supporting by not supporting

Line 230: what is a detectability pattern? Please rephrase

Line 236: I would suggest you used a word different than “wilder”. Maybe simply refer to a pattern more similar to that commonly seen on natural environments?

Line 238: “a rapid behavioural response to adapt to the new environmental conditions”. In my humble opinion, there is no need to talk about adaptation here and it is in fact confusing to do so. Why not simply use to better adjust to the new conditions?

Line 284: in line with previous comments, adaptation will only happen across generations, please rephrase. In my modest opinion, it is OK to say that a behavior is presumably ‘adaptive’, but not to say that birds are adapting their behavior in a temporal framework of days, as is written here.

Line 285-286: Same comment as before regarding adaptation.

Some of the species for which this study finds an increase in occurrence seem to be largely dependent on human-derived food (e.g. feral pigeons, doves, parakeets). One wonders whether it is possible that increased occurrence was due to increased movement rates of animals that are in worse body condition and therefore more motivated to actively search for food?

The final paragraph in some places seems to lack focus. For instance:

Line 313: importance for what? Please specify.

Line 318: bird behavior is altered as compared to what? I guess my point here is that it seems incongruent to state that in ‘normal’ conditions something is altered.

Line 322: Nosier should be noisier

Decision letter (RSPB-2020-2513.R0)

26-Nov-2020

Dear Dr Gordo:

Your manuscript has now been peer reviewed and the reviews have been assessed by an Associate Editor. The reviewers’ comments (not including confidential comments to the Editor) and the comments from the Associate Editor are included at the end of this email for your reference. As you will see, the reviewers and the Editors have raised some concerns with your manuscript and we would like to invite you to revise your manuscript to address them.

When submitting your revision please upload a file under "Response to Referees" - in the "File Upload" section. This should document, point by point, how you have responded to the reviewers’ and Editors’ comments, and the adjustments you have made to the manuscript. We require a copy of the manuscript with revisions made since the previous version marked as ‘tracked changes’ to be included in the ‘response to referees’ document.

Research ethics:

Use of animals and field studies:

It is a condition of publication that you make available the data and research materials supporting the results in the article. Please see our Data Sharing Policies (<https://royalsociety.org/journals/authors/author-guidelines/#data>). Datasets should be deposited in an appropriate publicly available repository and details of the associated accession number, link or DOI to the datasets must be included in the Data Accessibility section of the article (<https://royalsociety.org/journals/ethics-policies/data-sharing-mining/>). Reference(s) to datasets should also be included in the reference list of the article with DOIs (where available).

Please submit a copy of your revised paper within three weeks. If we do not hear from you within this time your manuscript will be rejected. If you are unable to meet this deadline please let us know as soon as possible, as we may be able to grant a short extension.

Best wishes,
Dr Maurine Neiman
mailto: proceedingsb@royalsociety.org

Associate Editor
Comments to Author:

Both referees were overall enthusiastic about the paper, and both agree that the dataset contribute to our understanding of the covid lockdown as a "natural experiment". At the same time, they raised a series of statistical and analytical queries that ought to be clarified - I will be looking forward to reading detailed explanations on how the authors addressed the queries raised by the referees.

Reviewer(s)' Comments to Author:

Referee: 1

Comments to the Author(s)

The paper tests the effects of COVID-19 lockdown on bird activity in NE Spain. This is interesting because it takes profit of an unrepeatable "natural experiment" to test popular ideas on the effect of the lockdown on nature. The data is very robust, since it includes 16 different species over a very large area including many cities and towns. The paper is also interesting because allows to better understand patterns of bird daily activity in urban environments.

The paper is also original in using occupancy models with variable "observer" as if it was a "square", allowing to quantify detectability. However, I wonder to which point the higher human activity in the first noisy hours of the day in "normal" urban days may reduce detectability and hence to provide a picture of lower activity at these hours, compared with more peaceful early hours in locked urban habitats. I assume that number of observers in 2020 was higher than in the unlocked years, so that detectability within the same observer could not be compared among years. Hence, some additional discussion or explanations would be appreciated on this point.

Given that lockdown was so sudden, could it be possible to check how many days could be necessary for birds to adapt their singing behaviour to the new situation?

Authors (lines 280-281) say that birds could assimilate the lockdown as a very long and especially peaceful weekend. Does it mean that they could change their singing behaviour one day to another? Some data and discussion on this would be interesting.

I have also a few minor comments:

Lines 48-49: In order to support your statement "However, some species are able to overcome these challenges and thrive in urban environments [4,8,13]", you cite papers:

4. McDonnell MJ, Hahs AK. 2015 Adaptation and adaptedness of organisms to urban environments.

351 *Annu. Rev. Ecol. Evol. S.* 46, 261-280.

8. Slabbekoorn H, Ripmeester EA. 2007 Birdsong and anthropogenic noise: implications and applications for conservation. *Mol. Ecol.* 17, 72-83

13. Bradley CA, Altizer S. 2007 Urbanization and the ecology of wildlife diseases. *Trends Ecol. Evol.* 22, 95-102

Although paper 4 is right, since it is general and reviews on the topic, the other two are quite specific. I think that instead of these two I would advice you to cite: (Johnson and Munshi-South 2017; Szulkin et al. 2020).

In lines 73-76, authors cite paper 32 to back up the idea of some that nature is getting back its space, but I think they wanted to cite paper 35. This happens again in line 224. Please check.

In figure 2, authors display three lines; one is red and another one green. I would like to point out that about 8% of men are colour blinded, which means that cannot distinguish between these two lines. Following advice from many different sources (eg. (Allred et al. 2014) I would encourage authors to change the colours of their figure.

References

- Allred SC, Schreiner WJ, Smithies O (2014) Colour blindness: still too many red-green figures. *Nature* 510:340. <https://doi.org/10.1038/510340e>
- Johnson MTJ, Munshi-South J (2017) Evolution of life in urban environments. *Science* 358:eaam8327. <https://doi.org/10.1126/science.aam8327>
- Szulkin M, Munshi-South J, Charmantier A (eds) (2020) *Urban evolutionary biology: Na. Munshi-South*. Oxford University Press, New York

Referee: 2

Comments to the Author(s)

This is a clever, straightforward study that uses data from a citizen science program to investigate whether reduced human activity in the context of a lockdown due to Covid-19 in a region of NE Spain led to shifts in bird activity patterns. The authors also investigate if detectability of some species changed throughout the day presumably due to associated rapid environmental changes such as noise or traffic reduction. Their main finding is that detectability of several bird species in urban settings at dawn increased by an average 27%, a pattern that may be seen as a convergence with activity patterns normally detected in non-urban environments.

This is a timely paper because it addresses a question that I assume numerous naturalists worldwide have been asking themselves. However, I have several questions that I would like to see clarified before I can recommend the paper for publication.

Gral comments:

My main concern is about the statistical consequences of comparing data from a single year (2020) to a pooled dataset of four previous years (2016-2019). The question is whether results would hold or, instead, whether among-year variation would erode significance in statistical results. I understand the authors possibly pooled together data from 2016-19 in order to have a sample size that was comparable to that of 2020 (which has altogether more bird checklists but a much lower number of sites) but it is possible such integration increased the chances of detecting a spurious statistical difference that would not be detected in comparisons year-by-year. It is important that the authors at least acknowledge the possible implications of this.

I wonder if multiple testing should be applied given that the authors are testing one independent hypothesis for each species. In the absence of multiple testing and taking a level of significance of $p=0.05$ when analyzing 16 species independently one would expect to detect, by chance, significant differences for at least one of these species.

Apart from the case of *C. carduelis*, an increase in the probability of occurrence was found for species that may be culled by local administrations like feral *C. livia* or the invasive *M. monachus*. Could the authors discuss if the lack of culling activities during lockdown could explain part of this variation? Finally, given recent range expansion of *S. decaocto* throughout Europe, one wonders whether there is some general trend of increased detection of this species

throughout the years that could explain an increase in sightings when comparing 2020 to the previous four years?

Figure 1: As stated above, a most valuable comparison—and the plot I would really like to see—is a figure showing probability of occurrence and confidence intervals for every year (i.e. 2016 through 2020) that showed evidence for an increase in 2020. This would be the most fair comparison instead of using the average for the previous four years. Indeed, the same comment applies to Figure 3.

Specific comments:

-Author contributions:

The way this is currently written, it seems like two of the four authors have not contributed to any relevant stage in this research. Please double check this.

-Abstract:

In general, it appears to me there is an incorrect use of 'adaptation' throughout the ms. The paper would do equally fine using alternative wording and the authors will prevent themselves from criticism by evolutionary ecologists. For example:

Lines 18-19: I would suggest to revisit this sentence given that 'mechanisms of adaptation' are not what is reported in this study which deals with changes in occurrence and daily patterns of activity.

Line 29: Same comment as above. Please consider editing the sentence "high behavioural plasticity to rapidly adapt to novel environmental conditions..."

-Introduction:

Line 53: Rephrase: "However, the mechanisms underlying the differences between urban and non-urban dwellers remain largely unknown"

Line 71: rephrase, less active cities?

Line 73: media and internet are not necessarily different things: e.g. social media commonly uses the network

Line 77: why is it relevant these are unexpected? By whom? The Intro could do as well with less subjective writing.

Line 87 and throughout the ms: please double check that 'quitter' is written instead of 'quieter'.

Line 102: "Moreover, as the masking effect of human acoustic contamination mostly disappeared, we expected an increase in singing activity...". I wonder if it would also be a valid alternative hypothesis to predict the opposite: if they communicate so well they might need less time to do the job.

Line 122: "to continue complete checklists (checklist with all identified species)". It might be useful to provide here in the Intro—apart from the Methods section—a very brief specification of what a complete checklist is?

Line 127-128: The authors state all observations were made from private homes. Is this the same as saying all these observations corresponded to urban habitats? In case there are people providing data who live in—or next to—natural areas, should there be an additional category of observations classified as 'lockdown non-urban'?

-Results:

Line 197: What is this "however" referring to?

Line 200: Excepting? Please reword.

Line 206-207: “In fact, the pattern of detectability along the day in the lockdown group resembled more to the non-urban pattern than to the urban pattern in many species” □ is there a formal way to provide a statistical test for this?

Line 211-214: I had a hard time understanding this sentence: what do “groups” refer to here? This is not properly indicated in Table S4 either.

Line 214: Please specify here what “time effect” refers to here.

-Discussion:

Line 223: please replace non-supporting by not supporting

Line 230: what is a detectability pattern? Please rephrase

Line 236: I would suggest you used a word different than “wilder”. Maybe simply refer to a pattern more similar to that commonly seen on natural environments?

Line 238: “a rapid behavioural response to adapt to the new environmental conditions”. In my humble opinion, there is no need to talk about adaptation here and it is in fact confusing to do so. Why not simply use to better adjust to the new conditions?

Line 284: in line with previous comments, adaptation will only happen across generations, please rephrase. In my modest opinion, it is OK to say that a behavior is presumably ‘adaptive’, but not to say that birds are adapting their behavior in a temporal framework of days, as is written here.

Line 285-286: Same comment as before regarding adaptation.

Some of the species for which this study finds an increase in occurrence seem to be largely dependent on human-derived food (e.g. feral pigeons, doves, parakeets). One wonders whether it is possible that increased occurrence was due to increased movement rates of animals that are in worse body condition and therefore more motivated to actively search for food?

The final paragraph in some places seems to lack focus. For instance:

Line 313: importance for what? Please specify.

Line 318: bird behavior is altered as compared to what? I guess my point here is that it seems incongruent to state that in ‘normal’ conditions something is altered.

Line 322: Nosier should be noisier

Author's Response to Decision Letter for (RSPB-2020-2513.R0)

See Appendix A.

RSPB-2020-2513.R1 (Revision)

Review form: Reviewer 2

Recommendation

Accept with minor revision (please list in comments)

Scientific importance: Is the manuscript an original and important contribution to its field?

Good

General interest: Is the paper of sufficient general interest?

Excellent

Quality of the paper: Is the overall quality of the paper suitable?

Good

Is the length of the paper justified?

Yes

Should the paper be seen by a specialist statistical reviewer?

No

Do you have any concerns about statistical analyses in this paper? If so, please specify them explicitly in your report.

Yes

It is a condition of publication that authors make their supporting data, code and materials available - either as supplementary material or hosted in an external repository. Please rate, if applicable, the supporting data on the following criteria.

Is it accessible?

N/A

Is it clear?

N/A

Is it adequate?

N/A

Do you have any ethical concerns with this paper?

No

Comments to the Author

I have now read the response to the editor and reviewers as well as the new version of this ms. I do not see new or additional issues with this article than I saw in my first review and I still believe it would be an interesting paper to be published given interest that the responses of animals to lockdown has brought not only to scientists and naturalists but also to the general public.

However, I was in general a bit disappointed by how the authors addressed some of the comments raised in the previous review. After reading the authors response, I was left with the impression they could have put a bit more effort to address some of these issues (or at least simply acknowledge them in the text).

-The most important issue I brought up in my first review was re. the statistical treatment of the data. I was concerned that comparing averages from data from a single year to data pooled from 5 years could result in inflated chances to detect average differences as compared with a year-by-year comparison. I realized the authors probably did that to have similar sample sizes (there were many more observations in 2020). I understand the peculiarities of the data collected for this study and this is why I did not ask the authors to repeat analyses doing year-by-year comparisons. However, I asked the authors to explicitly acknowledge this limitation, maybe in the Discussion section. Their first paragraph response to this question: "RESPONSE: Thanks for your comment. Interestingly, when we planned the analyses, we had the opposite thought. One may think: ok, let's compare 2020 only to 2019. However, in such pairwise comparison, it would be impossible to know whether differences are caused by conditions in 2020 or in 2019. A similar reasoning could be done for a comparison e.g. between 2020 and 2018. We pooled data from several years to have a representative baseline level or prior conditions to 2020. By pooling several years together, we avoided the potential peculiarities of each year (i.e., noise). Frankly, we

think that there are more chances to get spurious differences by comparing year to year than by comparing one year against an average. In a recent paper, Derryberry et al. (2020) used pooled data from two previous years (2015-16) to make comparison with the data collected during the 2020. We firmly believe that to pool years smooths interannual variability and improves comparisons." I understand these arguments but remain unconvinced that the argument is statistically sound. In the second paragraph of their response to this question, they state this was in fact a sample size limitation. I believe this is the correct answer, this is fine, I just felt it was important to highlight this limitation clearly in the ms. Regarding this same question, the authors give a more humble response to my comment about Figure 1 stating that year-by-year comparisons were not possible due to limited sample sizes.

Apart from this, I believe this is a relevant, timely study that will certainly be of interest to the scientific community.

Decision letter (RSPB-2020-2513.R1)

08-Jan-2021

Dear Dr Gordo:

Your manuscript has now been peer reviewed and the reviews have been assessed by an Associate Editor. The reviewers' comments (not including confidential comments to the Editor) and the comments from the Associate Editor are included at the end of this email for your reference. As you will see, the reviewers and the Editors have raised some concerns with your manuscript and we would like to invite you to revise your manuscript to address them.

Research ethics:

Use of animals and field studies:

It is a condition of publication that you make available the data and research materials supporting the results in the article (<https://royalsociety.org/journals/authors/author-guidelines/#data>). Datasets should be deposited in an appropriate publicly available repository and details of the associated accession number, link or DOI to the datasets must be included in the Data Accessibility section of the article (<https://royalsociety.org/journals/ethics-policies/data-sharing-mining/>). Reference(s) to datasets should also be included in the reference list of the article with DOIs (where available).

Please submit a copy of your revised paper within three weeks. If we do not hear from you within this time your manuscript will be rejected. If you are unable to meet this deadline please let us know as soon as possible, as we may be able to grant a short extension.

Best wishes,

Dr Maurine Neiman

Associate Editor

Board Member: 1

Comments to Author:

The authors have addressed the reviewer's comments. However, a few points are still outstanding:

- please address specifically in the discussion the point of the reviewer on year by year vs pooling argument

- I find the argument "most of our study sites are medium and small sized towns, where human derived food is hardly available." surprising, as it is not unusual in such towns to offer anthropogenic food

- One important point in the study remains unaddressed (it was not pointed out earlier), namely that at the moment, changes of detectability patterns, which you attribute to changes in avian behaviour, are in fact very difficult to distinguish from possible temporal shifts in human observation times. This is very loosely pointed out at the end of the discussion, but I would say it is very important to explicitly point out / discuss that changes in detectability can be caused by bias stemming from a change in the hours of work and leisure of confined observers (humans). I understand it will not be possible to disentangle this fully, but it is vital that this concern is addressed explicitly in the discussion and in the abstract.

Reviewer(s)' Comments to Author:

Referee: 2

Comments to the Author(s)

I have now read the response to the editor and reviewers as well as the new version of this ms. I do not see new or additional issues with this article than I saw in my first review and I still believe it would be an interesting paper to be published given interest that the responses of animals to lockdown has brought not only to scientists and naturalists but also to the general public.

However, I was in general a bit disappointed by how the authors addressed some of the comments raised in the previous review. After reading the authors response, I was left with the impression they could have put a bit more effort to address some of these issues (or at least simply acknowledge them in the text).

-The most important issue I brought up in my first review was re. the statistical treatment of the data. I was concerned that comparing averages from data from a single year to data pooled from 5 years could result in inflated chances to detect average differences as compared with a year-by-year comparison. I realized the authors probably did that to have similar sample sizes (there were many more observations in 2020). I understand the peculiarities of the data collected for this study and this is why I did not ask the authors to repeat analyses doing year-by-year comparisons. However, I asked the authors to explicitly acknowledge this limitation, maybe in the Discussion section. Their first paragraph response to this question: "RESPONSE: Thanks for your comment. Interestingly, when we planned the analyses, we had the opposite thought. One may think: ok, let's compare 2020 only to 2019. However, in such pairwise comparison, it would be impossible to know whether differences are caused by conditions in 2020 or in 2019. A similar reasoning could be done for a comparison e.g. between 2020 and 2018. We pooled data from several years to have a representative baseline level or prior conditions to 2020. By pooling several years together, we avoided the potential peculiarities of each year (i.e., noise). Frankly, we think that there are more chances to get spurious differences by comparing year to year than by comparing one year against an average. In a recent paper, Derryberry et al. (2020) used pooled data from two previous years (2015-16) to make comparison with the data collected during the

2020. We firmly believe that to pool years smooths interannual variability and improves comparisons." I understand these arguments but remain unconvinced that the argument is statistically sound. In the second paragraph of their response to this question, they state this was in fact a sample size limitation. I believe this is the correct answer, this is fine, I just felt it was important to highlight this limitation clearly in the ms. Regarding this same question, the authors give a more humble response to my comment about Figure 1 stating that year-by-year comparisons were not possible due to limited sample sizes.

Apart from this, I believe this is a relevant, timely study that will certainly be of interest to the scientific community.

Author's Response to Decision Letter for (RSPB-2020-2513.R1)

See Appendix B.

Decision letter (RSPB-2020-2513.R2)

29-Jan-2021

Dear Dr Gordo:

Your manuscript has now been peer reviewed and the reviews have been assessed by an Associate Editor. The reviewers' comments (not including confidential comments to the Editor) and the comments from the Associate Editor are included at the end of this email for your reference. As you will see, the reviewers and the Editors have raised some concerns with your manuscript and we would like to invite you to revise your manuscript to address them. You also need to ensure that you enclose a readily accessible file that shows track changes or something similar

Research ethics:

Use of animals and field studies:

It is a condition of publication that you make available the data and research materials supporting the results in the article (<https://royalsociety.org/journals/authors/author-guidelines/#data>). Datasets should be deposited in an appropriate publicly available repository and details of the associated accession number, link or DOI to the datasets must be included in the Data Accessibility section of the article (<https://royalsociety.org/journals/ethics-policies/data-sharing-mining/>). Reference(s) to datasets should also be included in the reference list of the article with DOIs (where available).

Please submit a copy of your revised paper within three weeks. If we do not hear from you within this time your manuscript will be rejected. If you are unable to meet this deadline please let us know as soon as possible, as we may be able to grant a short extension.

Best wishes,
Dr Maurine Neiman
Editor, Proceedings B
mailto: proceedingsb@royalsociety.org

Associate Editor
Board Member
Comments to Author:

Unless I have missed a document on manuscript central (it is sometimes hard to navigate), I can neither find a document addressing the specific comments that were raised after the first revision went it, nor a document with tracked changes so I see how the authors modified the revised manuscript.

It should be the authors responsibility to explicitly address editorial concerns. Unless I have not identified such document where the authors do explicitly show how they modified the manuscript (either by specifically replying to the points below, or by submitting a manuscript with tracked changes), I cannot assess this second revision.

Associate Editor
Board Member: 1
Comments to Author:

The authors have addressed the reviewer's comments. However, a few points are still outstanding:

- please address specifically in the discussion the point of the reviewer on year by year vs pooling argument

- I find the argument "most of our study sites are medium and small sized towns, where human derived food is hardly available." surprising, as it is not unusual in such towns to offer anthropogenic food

- One important point in the study remains unaddressed (it was not pointed out earlier), namely that at the moment, changes of detectability patterns, which you attribute to changes in avian behaviour, are in fact very difficult to distinguish from possible temporal shifts in human observation times. This is very loosely pointed out at the end of the discussion, but I would say it is very important to explicitly point out / discuss that changes in detectability can be caused by bias stemming from a change in the hours of work and leisure of confined observers (humans). I understand it will not be possible to disentangle this fully, but it is vital that this concern is addressed explicitly in the discussion and in the abstract.

Author's Response to Decision Letter for (RSPB-2020-2513.R2)

See Appendix C.

Decision letter (RSPB-2020-2513.R3)

11-Feb-2021

Dear Dr Gordo

I am pleased to inform you that your Review manuscript RSPB-2020-2513.R3 entitled "Rapid behavioural response of urban birds to COVID-19 lockdown" has been accepted for publication in Proceedings B.

The referee(s) do not recommend any further changes. Therefore, please proof-read your manuscript carefully and upload your final files for publication. Because the schedule for publication is very tight, it is a condition of publication that you submit the revised version of your manuscript within 7 days. If you do not think you will be able to meet this date please let me know immediately.

To upload your manuscript, log into <http://mc.manuscriptcentral.com/prsb> and enter your Author Centre, where you will find your manuscript title listed under "Manuscripts with Decisions." Under "Actions," click on "Create a Revision." Your manuscript number has been appended to denote a revision.

You will be unable to make your revisions on the originally submitted version of the manuscript. Instead, upload a new version through your Author Centre.

- 1) A text file of the manuscript (doc, txt, rtf or tex), including the references, tables (including captions) and figure captions. Please remove any tracked changes from the text before submission. PDF files are not an accepted format for the "Main Document".
- 2) A separate electronic file of each figure (tiff, EPS or print-quality PDF preferred). The format should be produced directly from original creation package, or original software format. Please note that PowerPoint files are not accepted.
- 3) Electronic supplementary material: this should be contained in a separate file from the main text and the file name should contain the author's name and journal name, e.g. `authorname_procb_ESM_figures.pdf`

All supplementary materials accompanying an accepted article will be treated as in their final form. They will be published alongside the paper on the journal website and posted on the online figshare repository. Files on figshare will be made available approximately one week before the accompanying article so that the supplementary material can be attributed a unique DOI. Please see: <https://royalsociety.org/journals/authors/author-guidelines/>

4) Data-Sharing and data citation

It is a condition of publication that data supporting your paper are made available. Data should be made available either in the electronic supplementary material or through an appropriate repository. Details of how to access data should be included in your paper. Please see <https://royalsociety.org/journals/ethics-policies/data-sharing-mining/> for more details.

<http://datadryad.org/submit?journalID=RSPB&manu=RSPB-2020-2513.R3> which will take you to your unique entry in the Dryad repository.

Once again, thank you for submitting your manuscript to Proceedings B and I look forward to receiving your final version. If you have any questions at all, please do not hesitate to get in touch.

Sincerely,
Dr Maurine Neiman
Editor, Proceedings B
<mailto:proceedingsb@royalsociety.org>

Associate Editor
Comments to Author:

Please add the analysis of change in detectability patterns as supplementary information, and explicitly refer to it in the body of the manuscript.

Decision letter (RSPB-2020-2513.R4)

16-Feb-2021

Dear Dr Gordo

I am pleased to inform you that your manuscript entitled "Rapid behavioural response of urban birds to COVID-19 lockdown" has been accepted for publication in Proceedings B.

Open Access

Your paper will be published with free Open Access as it deals with Covid-19, as part of our Covid-19 initiative.

Sincerely,
Editor, Proceedings B
mailto: proceedingsb@royalsociety.org

Appendix A

ASSOCIATE EDITOR

Both referees were overall enthusiastic about the paper, and both agree that the dataset contribute to our understanding of the covid lockdown as a "natural experiment". At the same time, they raised a series of statistical and analytical queries that ought to be clarified - I will be looking forward to reading detailed explanations on how the authors addressed the queries raised by the referees.

RESPONSE: We have addressed all the reviewers' concerns. Most of their requested clarifications or modifications have been fulfilled by rewriting some parts of the text or by adding some sentences. In our opinion, it is important to highlight that no major changes were requested or necessary. Therefore, we hope that our new version of the manuscript will be fully satisfactory now.

The reviewers provided valuable comments and suggestions that greatly improved our study. We thank them for an excellent review task and for their favourable reception of our study.

REFeree: 1

The paper tests the effects of COVID-19 lockdown on bird activity in NE Spain. This is interesting because it takes profit of an unrepeatable “natural experiment” to test popular ideas on the effect of the lockdown on nature. The data is very robust, since it includes 16 different species over a very large area including many cities and towns. The paper is also interesting because allows to better understand patterns of bird daily activity in urban environments.

The paper is also original in using occupancy models with variable “observer” as if it was a “square”, allowing to quantify detectability. However, I wonder to which point the higher human activity in the first noisy hours of the day in “normal” urban days may reduce detectability and hence to provide a picture of lower activity at these hours, compared with more peaceful early hours in locked urban habitats. I assume that number of observers in 2020 was higher than in the unlocked years, so that detectability within the same observer could not be compared among years. Hence, some additional discussion or explanations would be appreciated on this point.

RESPONSE: Thanks for your comment. We fully agree that we cannot discern the causal mechanisms for a reduction in early morning detectability during the lockdown period compared to the “normal” conditions. In fact, detectability can be modified both by a true behavioural change of birds and by a change in sampling conditions. The latter option was already discussed in a paragraph (L315-329) of the ms. However, as we pointed there, improved conditions for bird watching (e.g., reduced noise levels and pedestrians’ disturbance) during the lockdown were heavily constrained by the fact that observers were forced to remain at home. They could note down only what they saw from their balconies, yards or rooftops. This constrained conditions are well supported by the slower rate of species discovery during the lockdown (see Fig. S4 and Table S5) in most species.

There are two additional results supporting the view that that observed changes in detectability are reflecting, at least in part, a real behavioural response of birds to human lockdown. First, detectability change was a generalized response in almost all the studied species (see Fig. 3). Many of them are detected primarily by the songs of singing males, but some others are not, such as the magpie, the monk parakeet or the yellow-legged gull. Therefore, not all the observed changes can be simply related to an improved acoustic space facilitating bird sound detection. Second, and more interesting, the arising detection pattern during early morning resembles to the one observed in non-urban (=natural or rural) areas. In non-urban areas, there are no problems of detectability caused by anthropogenic noise or disturbance, as these areas are under a much lower human presence influence. Therefore, the non-urban pattern of detectability could be interpreted as showing the “pristine” behaviour of birds. For this reason, we included non-urban sites, as they serve as a control. Under “natural” conditions, birds show a peak of detectability in early morning. This can be attributed only to a greater activity (e.g., more singing). For this reason, when a very similar morning pattern of detectability was detected during the lockdown in the urban areas, we suggested that this can be interpreted as strong evidence of true behavioural change.

Yes, your assumption is right: During 2020, we gathered more checklists in urban areas than in previous years. This was due to the success of our project “#Istayathome”, by which observers kept their birding activities even during the lockdown. However, sample size alone cannot affect

detectability patterns. It can only affect the uncertainty of the parameters estimated in the models. For this reason, non-urban data showed narrower CI than the other groups, as we had much more data from non-urban than from urban areas.

Detectability was estimated within the same observer and site, as our sampling unit was the combination of observer+site, instead of simply site, as usually applied in previous studies using occupancy models (as the reviewer pointed). Our approach allowed changes in detectability within a site due to differences among observers to be removed effectively. This approach actually greatly enhanced the comparability among years, as variability between low- vs high-profile birdwatchers would not be affecting detectability estimates. In sum, the pool of observers involved in our citizen science project is the same before and during the covid-19 crisis, so any observer's effect is unlike.

Given that lockdown was so sudden, could it be possible to check how many days could be necessary for birds to adapt their singing behaviour to the new situation?

RESPONSE: This is a very interesting question! Unfortunately, we cannot test it for one main reason: sample size. Hierarchical occupancy models are quite demanding with the sample size. They provide robust estimates only with a large number of sites and replicates per site. If we split our lockdown dataset into shorter periods (e.g. days or weeks), samples become too small. To be honest, in a preliminary analysis, we split lockdown data into two groups: one for the first fortnight (second half of March) and another for the second one (first half of April). During the second one, lockdown conditions for people were more restrictive and the shutdown of the country was almost complete. As you can see in Fig. S1, many outdoor activities were even lower in the first half of April than in the second half of March. We hypothesized that lockdown effects should be more patent in April than in March. However, we did not see any strong difference between the two periods. Both showed similar patterns to the ones showed here for the lockdown group. Therefore, there were no objective reasons to split the lockdown data. Both groups showed greater uncertainty, as a consequence of reduced sample size. In fact, for some species we had some convergence issues for the model (i.e., model estimates were unreliable). In addition, by using 4 instead of 3 groups, interpretation and visualization of the results became more difficult (e.g., models had 7 parameters more). Computing time was also more demanding. In sum, we did not see any benefit from this approach and, for this reason, we opted for a simpler one with a single lockdown group, as we show in the manuscript.

Another potential issue is phenology. Lockdown happened in the middle of the reproductive season. Birds may show differences in singing behaviour related to date instead to lockdown conditions. For this reason, we excluded common migratory species in urban areas, as the house martin *Delichon urbicum*. They could be more detectable in mid-April than in mid-March just because more individuals arrived to their colonies late in the spring. A similar process happens for singing males of resident species. Singing males can be more (or less) detectable in mid-April because there is more (or less) reproductive activity later in the season. Date and lockdown day covary and their effects are impossible to taste apart. For this reason, we did not include sampling date as a covariate in the detectability part of the models.

In sum, although the reviewer question is really interesting and we in fact thought about it during ms preparation, we could not consider this question due to the limitations of our dataset. Covid-19 lockdown was so sudden that we could not plan properly sampling to address this question.

Authors (lines 280-281) say that birds could assimilate the lockdown as a very long and especially peaceful weekend. Does it mean that they could change their singing behaviour one day to another? Some data and discussion on this would be interesting.

RESPONSE: Our comparison was rather metaphoric in the sense that urban birds are already used to experience relatively peaceful days: weekends. Several previous studies have demonstrated that birds are in fact able to change their behaviour from one day to another as they show different routines between week and weekend days (e.g., see Bautista et al. 2004, Díaz et al. 2011). As we discussed in our previous reply, we cannot know for certain how long it took for the birds to change their behaviour in response to the lockdown. Probably, it took very little time. Our study lasted only 1 month and we already found patent differences. Therefore, behavioural adjustment happened necessarily in less than four weeks. However, this is not surprising at all. As pointed before, several studies demonstrated this ability of birds of changing their behaviour from one day to another, so many of them adjusted probably to the new covid-19 lockdown conditions almost immediately.

I have also a few minor comments:

Lines 48-49: In order to support your statement “However, some species are able to overcome these challenges and thrive in urban environments [4,8,13]”, you cite papers:

4. McDonnell MJ, Hahs AK. 2015 Adaptation and adaptedness of organisms to urban environments. *Annu. Rev. Ecol. Evol. S.* 46, 261–280.
8. Slabbekoorn H, Ripmeester EA. 2007 Birdsong and anthropogenic noise: implications and applications for conservation. *Mol. Ecol.* 17, 72–83
13. Bradley CA, Altizer S. 2007 Urbanization and the ecology of wildlife diseases. *Trends Ecol. Evol.* 22, 95–102

Although paper 4 is right, since it is general and reviews on the topic, the other two are quite specific. I think that instead of these two I would advice you to cite: (Johnson and Munshi-South 2017; Szulkin et al. 2020).

RESPONSE: Thanks for your advice. We have included the suggested references.

In lines 73-76, authors cite paper 32 to back up the idea of some that nature is getting back its space, but I think they wanted to cite paper 35. This happens again in line 224. Please check.

RESPONSE: Thanks for this remark. In fact, citation was wrong. We wanted to cite Manenti et al. (2020) instead of Pacifici et al. (2008).

In figure 2, authors display three lines; one is red and another one green. I would like to point out that about 8% of men are colour blinded, which means that cannot distinguish between these two lines. Following advice from many different sources (eg. (Allred et al. 2014) I would encourage authors to change the colours of their figure.

RESPONSE: Thanks for remembering us about this important issue! We have changed red and green by orange and blue, respectively, in all figures.

References

- Allred SC, Schreiner WJ, Smithies O (2014) Colour blindness: still too many red-green figures. *Nature* 510:340. <https://doi.org/10.1038/510340e>
- Johnson MTJ, Munshi-South J (2017) Evolution of life in urban environments. *Science* 358:eaam8327. <https://doi.org/10.1126/science.aam8327>
- Szulkin M, Munshi-South J, Charmantier A (eds) (2020) *Urban evolutionary biology*: Na. Munshi-South. Oxford University Press, New York

REFeree: 2

This is a clever, straightforward study that uses data from a citizen science program to investigate whether reduced human activity in the context of a lockdown due to Covid-19 in a region of NE Spain led to shifts in bird activity patterns. The authors also investigate if detectability of some species changed throughout the day presumably due to associated rapid environmental changes such as noise or traffic reduction. Their main finding is that detectability of several bird species in urban settings at dawn increased by an average 27%, a pattern that may be seen as a convergence with activity patterns normally detected in non-urban environments.

This is a timely paper because it addresses a question that I assume numerous naturalists worldwide have been asking themselves. However, I have several questions that I would like to see clarified before I can recommend the paper for publication.

Gral comments:

My main concern is about the statistical consequences of comparing data from a single year (2020) to a pooled dataset of four previous years (2016-2019). The question is whether results would hold or, instead, whether among-year variation would erode significance in statistical results. I understand the authors possibly pooled together data from 2016-19 in order to have a sample size that was comparable to that of 2020 (which has altogether more bird checklists but a much lower number of sites) but it is possible such integration increased the chances of detecting a spurious statistical difference that would not be detected in comparisons year-by-year. It is important that the authors at least acknowledge the possible implications of this.

RESPONSE: Thanks for your comment. Interestingly, when we planned the analyses, we had the opposite thought. One may think: ok, let's compare 2020 only to 2019. However, in such pairwise comparison, it would be impossible to know whether differences are caused by conditions in 2020 or in 2019. A similar reasoning could be done for a comparison e.g. between 2020 and 2018. We pooled data from several years to have a representative baseline level or prior conditions to 2020. By pooling several years together, we avoided the potential peculiarities of each year (i.e., noise). Frankly, we think that there are more chances to get spurious differences by comparing year to year than by comparing one year against an average. In a recent paper, Derryberry et al. (2020) used pooled data from two previous years (2015-16) to make comparison with the data collected during the 2020. We firmly believe that to pool years smooths interannual variability and improves comparisons.

Another important reason to merge several years was sample size. Previously to covid-19 pandemic, observers recorded few checklists in urban areas. People usually prefers natural sites for birdwatching. For this reason, we do not have enough data to run our models using data from single previous years to 2020. The actual problem for previous years was a deficit of replicated samples, which are essential for a robust estimation of detection probability. For this reason, we decided to combine the 5 previous years (2015-2019) to use them as baseline level for comparisons. Five years is a short frame to observe drastic distribution changes in our studied species, as most of them are well-established urban dwellers and have stable populations.

We have included a couple of sentences in the methods (L135-8) to justify our approach.

I wonder if multiple testing should be applied given that the authors are testing one independent hypothesis for each species. In the absence of multiple testing and taking a level of significance of $p=0.05$ when analyzing 16 species independently one would expect to detect, by chance, significant differences for at least one of these species.

RESPONSE: We agree with this comment, but only partially. The $p=0.05$ was used for convenience, as it is the most used cut line in scientific research. We are fully aware that this is simply an arbitrary threshold.

There were not 16 independent tests. There were lot of more. Note the large number of variables and interactions included in each model. They implied 21 p-values estimated for the parameters included in the model of species plus 3 p-values from the log-likelihood ratio tests. In total, there were 384 p-values. For this reason, we interpreted the results as a whole. We did not focus our attention in the significance of a particular variable in a particular species. Instead, we made an overall assessment of the effects looking, for instance, how many species showed significant effects for a certain variable and/or how many showed effects in agreement with our predictions.

In most cases, effects were strongly significant. For instance, hour effect was $p<0.00001$ in more than half of species. Thus, it is obvious that birds showed different detectability along the day. This is not a surprise at all. Except for a couple of species, all species showed also an effect hour*group well below the 0.05 arbitrary threshold ($p<0.005$), suggesting that in most cases the hour effect differed among groups. A similar reasoning could be applied for the occupancy part of the model. Most of the species did not differ significantly between the lockdown and the urban group. If one of those differences was spurious, then it would imply that lockdown and urban occurrence were even more similar. Our conclusion would be the same (or it would be even reinforced). However, in the detection part, effects were so marked that one or a few spurious effects would not affect the overall picture showed in our study. For these reasons, we think that multiple testing is not necessary.

Apart from the case of *C. carduelis*, an increase in the probability of occurrence was found for species that may be culled by local administrations like feral *C. livia* or the invasive *M. monachus*. Could the authors discuss if the lack of culling activities during lockdown could explain part of this variation? Finally, given recent range expansion of *S. decaocto* throughout Europe, one wonders whether there is some general trend of increased detection of this species throughout the years that could explain an increase in sightings when comparing 2020 to the previous four years?

RESPONSE: These are good remarks, which may apply for other countries or regions. However, they are really improbable or not feasible in our case. First, in the case of feral pigeons and invasive parakeets, local administrations have not carried out culling activities for many years now due to the strong social opposition against such managing practices. From a conservation point of view, this is especially serious in the case of parakeets, which are having some negative effects on native avifauna. Unfortunately, these birds have become too popular for people, i.e. public opinion is fairly favourable to the occurrence of Psittacidae in the cities, in spite of the scientific consensus of their harmful effects (competence for food and breeding places, potential disease transmission, crop damages, etc.; see e.g. Senar et al. 2016; Covas et al. 2017).

In the case of feral pigeons, there were initiatives to control their populations in the Barcelona city in the past, as their numbers may represent a public health issue. Once again, culling of pigeons was criticized by some social sectors and consequently, ruled out by authorities. In the last three years, bird feeders with food treated with contraceptive drugs have been used. Unfortunately, contraceptive feed has been useless (Senar et al. 2020). In any case, note that pigeon population is mainly managed in the city of Barcelona, which is just a city of our broad sample of urban areas of Catalonia.

Finally, in the case of the collared dove, this species is established in Catalonia since the end of the 70s (Pocino et al. 2005). Currently, it occupies >80% of the territory and only lacks in the most isolated parts of the Pyrenees (<http://www.sioc.cat/fitxa.php?sp=STRDEC>). The common breeding birds monitoring program of Catalonia (called SOCC) demonstrates a slight increase of 2% per year since 2003. SOCC is a standardized monitoring program of bird populations in Catalonia, similar to those conducted in the rest of European countries, and included in the PECBMS. Therefore, the occupation increase of 18% detected in our model cannot be explained by the species range expansion, which is negligible in Catalan urban areas.

As we suggested in the first paragraph of the discussion, the occurrence increase of these three species is probably due to a more samples in urban core areas. We have added some new sentences including previous explanations.

Figure 1: As stated above, a most valuable comparison—and the plot I would really like to see—is a figure showing probability of occurrence and confidence intervals for every year (i.e. 2016 through 2020) that showed evidence for an increase in 2020. This would be the most fair comparison instead of using the average for the previous four years. Indeed, the same comment applies to Figure 3.

RESPONSE: Unfortunately, as we explained previously, this is not feasible due to the limited sample size available annually between 2015 and 2019.

Specific comments:

-Author contributions:

The way this is currently written, it seems like two of the four authors have not contributed to any relevant stage in this research. Please double check this.

RESPONSE: LB and SH incorporated to this study in a late stage and provided essential guidance, advices and suggestions during the writing phase, which is indeed a very relevant phase.

-Abstract:

In general, it appears to me there is an incorrect use of ‘adaptation’ throughout the ms. The paper would do equally fine using alternative wording and the authors will prevent themselves from criticism by evolutionary ecologists. For example:

Lines 18-19: I would suggest to revisit this sentence given that ‘mechanisms of adaptation’ are not what is reported in this study which deals with changes in occurrence and daily patterns of activity.

Line 29: Same comment as above. Please consider editing the sentence “high behavioural plasticity to rapidly adapt to novel environmental conditions...”

RESPONSE: Thanks for this remark. We have made the necessary changes to avoid a wrong use of the term “adaptation” (L19, L29, L262, L310-11).

-Introduction:

Line 53: Rephrase: “However, the mechanisms underlying the differences between urban and non-urban dwellers remain largely unknown”

RESPONSE: Changed to: “However, little is known about the adaptive mechanisms allowing the differences observed between urban and non-urban dwellers”

Line 71: rephrase, less active cities?

RESPONSE: Changed to: “...to less crowded, noisy and polluted cities...”

Line 73: media and internet are not necessarily different things: e.g. social media commonly uses the network

RESPONSE: Reworded to: “...the media and social networks...”

Line 77: why is it relevant these are unexpected? By whom? The Intro could do as well with less subjective writing.

RESPONSE: We have changed unexpected by sudden. Anyway, we think that nobody expected the covid-19 pandemic and their consequences less than one year ago. Nobody could imagine in February 2020 that most countries would impose severe restrictions to human activities during the 2020 spring and summer (many of them still going on). In this sense, the covid-19 has been “unexpected”.

Line 87 and throughout the ms: please double check that ‘quitter’ is written instead of ‘quieter’.

RESPONSE: Thanks! Amended (L28, L85, L98, L317, L356).

Line 102: “Moreover, as the masking effect of human acoustic contamination mostly disappeared, we expected an increase in singing activity...”. I wonder if it would also be a valid alternative hypothesis to predict the opposite: if they communicate so well they might need less time to do the job.

RESPONSE: Yes, we agree with the reviewer opinion, but empirical evidences (e.g. see Díaz et al. (2011), Gil et al. (2015)), have demonstrated that birds sing more in silent than in noisy environments. Apparently, birds use as much time as possible for singing. More time singing may imply to attract more females (and potential copulas) or may imply a more efficient

territory defence. In any case, more singing activity is generally related to increased fitness and, for this reason, males invest as much time as possible singing. In noisy environments, singing becomes a futile activity. Males only pay its costs in terms of wasted time, invested energy and increased predation risk, as females cannot hear (or can hardly hear) them. Singing more time, louder and/or more complex songs does not make the difference anymore. In this scenario, birds reduce their singing activity. Therefore, when the masking effect of human acoustic contamination disappears, we expect “normal” levels of singing activity, which are higher.

Line 122: “to continue complete checklists (checklist with all identified species)” . It might be useful to provide here in the Intro—apart from the Methods section—a very brief specification of what a complete checklist is?

RESPONSE: We are sorry, but we do not understand this comment. L122 was in the methods section. We do not mention checklists in the intro.

Line 127-128: The authors state all observations were made from private homes. Is this the same as saying all these observations corresponded to urban habitats? In case there are people providing data who live in—or next to—natural areas, should there be an additional category of observations classified as ‘lockdown non-urban’?

RESPONSE: This is a good observation. In our study, private homes were in the overwhelming majority of cases in compact urban habitats. As we explain in the methods (L129), site categorization (urban vs non-urban) was done by GIS tools based on the land-use at the coordinates provided by the observers. Only four sites were placed on non-urban land uses in the lockdown group. They corresponded to observers from isolated houses in rural areas. Unfortunately, this is not enough to create a ‘lockdown non-urban’ group and consequently, we excluded these data from our study. However, we fully agree with the reviewer that this group would have been really interesting and the best possible control group. As we explained above, lockdown was so sudden that we were unable to plan properly sampling to address interesting questions such as this one.

-Results:

Line 197: What is this “however” referring to?

RESPONSE: Deleted.

Line 200: Excepting? Please reword.

RESPONSE: Changed by “Except for...”

Line 206-207: “In fact, the pattern of detectability along the day in the lockdown group resembled more to the non-urban pattern than to the urban pattern in many species” is there a formal way to provide a statistical test for this?

RESPONSE: This statement is based on a visual inspection of the fig. 2 and S3. As the hour effect was modelled by splines, there is no way to test differences among curves in an analogous way to a test of heterogeneity of slopes in a lineal regression or a postdoc test in a ANOVA. In fact, the hour effect was modelled using a GAM like approach. In table S4, we included the p-value for the interaction $f(\text{hour}) \cdot \text{group}$. This test allowed to confirm whether the three curves were equal or not. A posterior visual inspection of the plots allowed to determine where such differences arose.

Line 211-214: I had a hard time understanding this sentence: what do “groups” refer to here? This is not properly indicated in Table S4 either.

RESPONSE: “Groups” refer here, as in the rest of the ms, to the three groups of data: observations done during the 2020 lockdown, historical records from urban areas between 2015-19, and historical records from non-urban areas between 2015-19. This sentence as well as table S4 can be understood similarly to the previous reply. The effect of time was modelled by a general slope plus a particular slope for each group. This is what happens when one makes an interaction between a continuous (i.e., time) and categorical (i.e., group) variable (i.e., $\text{time} \cdot \text{group}$). In order to save df, the model takes one group as reference, fits the line, and estimates a parameter for it. Then, for the rest of groups, the model just adds a value to this reference slope and tests whether or not the added value is significantly different from 0. All these results are summarized in Table S5. As expected, sampling duration showed a significant positive effect in all species, at least, in one of the groups (i.e. in all species you can see one or more increasing lines in Fig. S4). The goldfinch, feral pigeon, and black redstart showed flat lines for the lockdown data. Slopes (parameters) were close to 0.5 in these three cases and none of them were indeed significant. (0.5 is equivalent to slope=0 in a typical lineal regression, but note that we are working in log scale). However, in the previously mentioned three species, slopes in the historical urban and non-urban groups were significantly different. This means that these slopes were indeed different from 0.5. In table S4, we are simply providing an overall test for the interaction $\text{time} \cdot \text{group}$, instead of the particular effects fitted to each group (table S5). This overall test informs whether or not fitting separate slopes for each group improves the model. In some species, such as the wood pigeon or the blue tit, this test is clearly far from significance. If you look at table S5, you will see that the slope for the historical urban and non-urban groups did not differ from the lockdown group in these two species. This result means that, under a longer sampling, we have more chances to find a wood pigeon or a blue tit. However, chances increase at the same rate with sampling time in the lockdown, historical urban and non-urban groups. If you look at fig. S4, you will see that slopes are quite similar (i.e. “parallel”).

We have rewritten that sentence as: Therefore, a certain increase of the sampling time implied a different increase in chances of detection in the lockdown, the historical urban and the historical non-urban groups for most species.

Line 214: Please specify here what “time effect” refers to here.

RESPONSE: We have specified that it is sampling time.

-Discussion:

Line 223: please replace non-supporting by not supporting

RESPONSE: Replaced.

Line 230: what is a detectability pattern? Please rephrase

RESPONSE: Rephrased as: "...detectability pattern throughout the day as a consequence..."

Line 236: I would suggest you used a word different than "wilder". Maybe simply refer to a pattern more similar to that commonly seen on natural environments?

RESPONSE: Rephrased as: "Overall, many species showed a pattern of detection during the lockdown in urban areas more similar to that commonly seen on natural environments."

Line 238: "a rapid behavioural response to adapt to the new environmental conditions". In my humble opinion, there is no need to talk about adaptation here and it is in fact confusing to do so. Why not simply use to better adjust to the new conditions?

RESPONSE: We have changed adapt by adjust.

Line 284: in line with previous comments, adaptation will only happen across generations, please rephrase. In my modest opinion, it is OK to say that a behavior is presumably 'adaptive', but not to say that birds are adapting their behavior in a temporal framework of days, as is written here.

RESPONSE: Thanks once again for your remarks on the right use of "adaptation". We have changed adapt by modify.

Line 285-286: Same comment as before regarding adaptation.

RESPONSE: We have replaced adapt by habituate.

Some of the species for which this study finds an increase in occurrence seem to be largely dependent on human-derived food (e.g. feral pigeons, doves, parakeets). One wonders whether it is possible that increased occurrence was due to increased movement rates of animals that are in worse body condition and therefore more motivated to actively search for food?

RESPONSE: This is a good hypothesis, but, to the best of our knowledge, the only species relying on human-derived food in our study area are feral pigeons, house sparrows and gulls. For instance, as we explained above, bird feeders with contraceptive seeds for the feral pigeons have been used in the last three years in Barcelona city. Both doves and parakeets are quite distrustful. They tolerate pretty well human presence, but they mainly rely on seeds and fruits from plants in yards and parks. For example, monk parakeets are a pest for crops near to Barcelona city (see Senar et al 2016). Maybe, this different behaviour of doves and parakeets is

related to a more recent occupation of urban areas than other bird species, as pigeons or sparrows. Finally, it is important to stress that all these comments about human-derived food for birds should affect mostly to Barcelona city (and perhaps a few other large cities of Catalonia). However, we had data for 149 urban sites during the lockdown and for 410 urban sites during the historical period 2015-2019. Therefore, most of our study sites are medium and small sized towns, where human derived food is hardly available.

The final paragraph in some places seems to lack focus. For instance:

Line 313: importance for what? Please specify.

RESPONSE: Rephrased as: "...about the importance of urban populations for bird conservation".

Line 318: bird behavior is altered as compared to what? I guess my point here is that it seems incongruent to state that in 'normal' conditions something is altered.

RESPONSE: This incongruence is just what we want to emphasize here. What we are used to call "normal conditions" are rather "abnormal conditions". Of course, life in urban environments would be "abnormal" by comparison with natural conditions. The lockdown experience has reminded us this fact, which is actually obvious. However, we are so used to live in crowded, noisy, polluted and "unnatural" cities that we are oblivious of such obviousness. In any case, we have rephrased the text as: "Under typical urban conditions, bird behaviour is apparently altered by our lifestyle and...".

Line 322: Nosier should be noisier

RESPONSE: Amended.

Appendix B

Associate Editor

The authors have addressed the reviewer's comments. However, a few points are still outstanding:

- please address specifically in the discussion the point of the reviewer on year by year vs pooling argument

RESPONSE: As requested by the reviewer 2, we have added several sentences at the end of the 2nd paragraph of the discussion (L261-8) to explain the potential consequences: “[...] *Although in urban areas the 2020 detectability pattern was patently different from the 2015-2019 baseline level, such differences could have been partially enhanced because of comparing a single year against several pooled years. Nevertheless, it would be necessary to know whether or not there is significant among years’ variability in bird daily routines. If that is the case, we would expect that behavioural responses to extreme events, such as the covid-19 lockdown, would fall out of the normal variability. Unfortunately, we could not properly explore the between years’ variation in detectability patterns due to sample size constrains. [...]*”. Furthermore, we pointed out this potential issue in the methods (L138): “[...] *COVID-19 pandemics, although we could not assess variability among years. [...]*”

- I find the argument “most of our study sites are medium and small sized towns, where human derived food is hardly available.” surprising, as it is not unusual in such towns to offer anthropogenic food

RESPONSE: We can understand the editor’s surprise, as people’s behaviour towards urban birds is probably quite different in Spain compared to other European or North American cities. First, when we talk about medium and small sized towns, maybe it is important to give you a reference numbers. 78.1% of Catalan municipalities have less than 5,000 inhabitants; more than a half are below 1,000.

As we explained in our previous reply letter, bird feeders in gardens and yards are still a very rare practice in our study region, especially in the countryside, where the overwhelming majority of these small towns/villages are located. There is no tradition and culture to actively feed wild birds. Actually, this can be also applied to the rest of Spain. Maybe, Iberian mild winters make unnecessary supplementary feeding for bird survival and, for this reason, feeders are not very popular. Or maybe, there are still more people interested to eat songbirds, instead of providing food to them. It is important to know our social context, which is (as stated above) probably quite different from other countries. Therefore, bird feeders are not an option for urban birds in Spain due to its scarcity.

Alternatively, urban birds could use litter bins as a food source. There, they may find human-derived food *sensu lato*. Once again, we think that this cannot be a relevant food source for urban birds. Litter bins are closed containers, which are collected every day. Thus, little or no waste is accessible to feral urban fauna by this means. However, open refuse dumps can play an important role as an infinite and predictable source of resources for several bird species, as gulls, storks, or raptors. There are only 29 large dumps in Catalonia, all of them located far from urban areas to avoid nuisance to people (for instance, only 5 within a 70km radius of Barcelona). Therefore, dumps cannot be considered a part of the urban landscape in Catalonia.

Furthermore, excepting gulls, dumps are not a suitable environment for the rest of the studied species.

Maybe the only source of human-derived food in cities worth to be mentioned is provided by retired people. Some of them have as a pastime to go to urban parks to feed bread to pigeons. Sometimes, sparrows or even parakeets may profit these situations to eat some breadcrumbs. However, this is a typical pastime in the most urbanized areas, such as Barcelona city, where feeding pigeons was also a popular attraction for tourists. However, this practice was banned some years ago in the city as part of the pigeon population control program.

In sum, as a rule, human-derived food is scarce in the urban areas of Catalonia. Even when food is available, it benefits only a few localized bird populations (e.g., pigeons in Barcelona parks). All potential sources cited before (feeders, bins, retired people) are more plausible to occur in large urban areas, as Barcelona and its metropolitan area. However, this area represents just a minor fraction of the urban areas of all Catalonia. Catalan urban areas are mainly represented by small villages and towns in the countryside. There, citizens are probably less interested on/friendly towards avian urban fauna than in big cities.

- One important point in the study remains unaddressed (it was not pointed out earlier), namely that at the moment, changes of detectability patterns, which you attribute to changes in avian behaviour, are in fact very difficult to distinguish from possible temporal shifts in human observation times. This is very loosely pointed out at the end of the discussion, but I would say it is very important to explicitly point out / discuss that changes in detectability can be caused by bias stemming from a change in the hours of work and leisure of confined observers (humans). I understand it will not be possible to disentangle this fully, but it is vital that this concern is addressed explicitly in the discussion and in the abstract.

RESPONSE: Thanks for your comment. As you say, we have been aware about this potential issue and we devoted the 6th paragraph of the discussion to discuss the pro and cons of this hypothesis. We fully agree that we cannot know what the cause is for the detectability change during the lockdown period. It could be because of a true behavioural change, an improvement in sampling conditions, or a combination of both. An improvement in sampling conditions does not seem probable, as the observers were strongly constrained in their surveys (they could watch birds only from home). In fact, the effect of sampling time showed a slower rate of discovery in all species during the lockdown year than during previous years. This provides evidence of a constrained ability of birders to find birds. As most of our volunteers have been engaged in our citizen science project since many years now, this result cannot be due to a lower profile of the observers during the lockdown period than before. Indeed, observers were the same people before and during the lockdown.

An inspection of the distribution of checklists along the day would not support the editor's hypothesis of the sampling bias. Remember that -1 is sunrise, 0 is midday, and +1 is sunset. The overall pattern is quite similar in the three groups. In all cases, most of checklists are carried out during the first half of the morning. There is another peak of observers' activity in the evening.

If we calculate the difference between the % of checklists in the lockdown and urban groups, we can make a closer examination of the editor’s hypothesis:

Positive values mean more % of checklists in the lockdown group, while negative values mean more % of checklists in the historical urban group. As you can see, differences are always pretty low. Average absolute difference is just 1.26%. There are larger differences for some hourly periods (e.g., at -0.7), but such differences are never systematic. For instance, there was a slightly higher % of checklists during the lockdown at sunrise (hour = -1), but in the period 2015-19 there were more checklist just before sunrise. Thus, differences move in an erratic fashion up and down around zero. From this pattern, it can be concluded that there were no temporal changes in the observation times.

Last but not least, detectability cannot be affected by sample size. The number of observers watching birds will determine the available sample size and consequently the error of the estimated parameters (i.e. their uncertainty). For this reason, error bands were wider in the lockdown and urban group compared to the non-urban group (see fig. 2). However, sample size should not affect the parameters’ magnitude, which is the truly relevant factor in this case. Detectability depends on birds’ behaviour as well as on environmental conditions that affect our ability to see/hear them. In general, one can assume confidently that nor the behaviour neither the environmental conditions can be affected by the number of birdwatchers traipsing the streets.

Referee: 2

I have now read the response to the editor and reviewers as well as the new version of this ms. I do not see new or additional issues with this article than I saw in my first review and I still believe it would be an interesting paper to be published given interest that the responses of animals to lockdown has brought not only to scientists and naturalists but also to the general public.

RESPONSE: Thank you very much for reviewing again our manuscript.

However, I was in general a bit disappointed by how the authors addressed some of the comments raised in the previous review. After reading the authors response, I was left with the impression they could have put a bit more effort to address some of these issues (or at least simply acknowledge them in the text).

RESPONSE: We are very sorry to hear this, as it was not our intention to leave any previous comments and concerns properly unaddressed. We really hope that with this new version, we make all the necessary changes in the manuscript according to the reviewer suggestions and comments.

The most important issue I brought up in my first review was re. the statistical treatment of the data. I was concerned that comparing averages from data from a single year to data pooled from 5 years could result in inflated chances to detect average differences as compared with a year-by-year comparison. I realized the authors probably did that to have similar sample sizes (there were many more observations in 2020). I understand the peculiarities of the data collected for this study and this is why I did not ask the authors to repeat analyses doing year-by-year comparisons. However, I asked the authors to explicitly acknowledge this limitation, maybe in the Discussion section. Their first paragraph response to this question: *“RESPONSE: Thanks for your comment. Interestingly, when we planned the analyses, we had the opposite thought. One may think: ok, let’s compare 2020 only to 2019. However, in such pairwise comparison, it would be impossible to know whether differences are caused by conditions in 2020 or in 2019. A similar reasoning could be done for a comparison e.g. between 2020 and 2018. We pooled data from several years to have a representative baseline level or prior conditions to 2020. By pooling several years together, we avoided the potential peculiarities of each year (i.e., noise). Frankly, we think that there are more chances to get spurious differences by comparing year to year than by comparing one year against an average. In a recent paper, Derryberry et al. (2020) used pooled data from two previous years (2015-16) to make comparison with the data collected during the 2020. We firmly believe that to pool years smooths interannual variability and improves comparisons.”* I understand these arguments but remain unconvinced that the argument is statistically sound. In the second paragraph of their response to this question, they state this was in fact a sample size limitation. I believe this is the correct answer, this is fine, I just felt it was important to highlight this limitation clearly in the ms. Regarding this same question, the authors give a more humble response to my comment about Figure 1 stating that year-by-year comparisons were not possible due to limited sample sizes.

RESPONSE: We have added several sentences at the end of the 2nd paragraph of the discussion (L261-8) to explain the potential consequences: “[...] Although in urban areas the

2020 detectability pattern was patently different from the 2015-2019 baseline level, such differences could have been partially enhanced because of comparing a single year against several pooled years. Nevertheless, it would be necessary to know whether or not there is significant among years' variability in bird daily routines. If that is the case, we would expect that behavioural responses to extreme events, such as the covid-19 lockdown, would fall out of the normal variability. Unfortunately, we could not properly explore the between years' variation in detectability patterns due to sample size constrains. [...]". Furthermore, we pointed out this potential issue in the methods (L138): “[...] COVID-19 pandemics, although we could not assess variability among years. [...]"

We really appreciated the reviewer's comment about this issue, and we will consider it in the future to improve our citizen science project (e.g. convincing the volunteers to make more surveys in urban areas).

Apart from this, I believe this is a relevant, timely study that will certainly be of interest to the scientific community.

RESPONSE: Thanks a lot for your task as a reviewer, and thanks for your positive feedback on our manuscript. Despite our different opinions on certain points, we really appreciate all of your comments, as well as those from the other reviewer and the editor. They have greatly improved our study. We apologize for any misunderstandings raised from our previous reply letter.

Appendix C

Associate Editor

The authors have addressed the reviewer's comments. However, a few points are still outstanding:

- please address specifically in the discussion the point of the reviewer on year by year vs pooling argument

RESPONSE: As requested by the reviewer 2, we have added several sentences at the end of the 2nd paragraph of the discussion (L261-8) to explain the potential consequences: “[...] *Although in urban areas the 2020 detectability pattern was patently different from the 2015-2019 baseline level, such differences could have been partially enhanced because of comparing a single year against several pooled years. Nevertheless, it would be necessary to know whether or not there is significant among years’ variability in bird daily routines. If that is the case, we would expect that behavioural responses to extreme events, such as the covid-19 lockdown, would fall out of the normal variability. Unfortunately, we could not properly explore the between years’ variation in detectability patterns due to sample size constrains. [...]*”. Furthermore, we pointed out this potential issue in the methods (L138): “[...] *COVID-19 pandemics, although we could not assess variability among years. [...]*”

- I find the argument “most of our study sites are medium and small sized towns, where human derived food is hardly available.” surprising, as it is not unusual in such towns to offer anthropogenic food

RESPONSE: We can understand the editor’s surprise, as people’s behaviour towards urban birds is probably quite different in Spain compared to other European or North American cities. First, when we talk about medium and small sized towns, maybe it is important to give you a reference numbers. 78.1% of Catalan municipalities have less than 5,000 inhabitants; more than a half are below 1,000.

As we explained in our previous reply letter, bird feeders in gardens and yards are still a very rare practice in our study region, especially in the countryside, where the overwhelming majority of these small towns/villages are located. There is no tradition and culture to actively feed wild birds. Actually, this can be also applied to the rest of Spain. Maybe, Iberian mild winters make unnecessary supplementary feeding for bird survival and, for this reason, feeders are not very popular. Or maybe, there are still more people interested to eat songbirds, instead of providing food to them. It is important to know our social context, which is (as stated above) probably quite different from other countries. Therefore, bird feeders are not an option for urban birds in Spain due to its scarcity.

Alternatively, urban birds could use litter bins as a food source. There, they may find human-derived food *sensu lato*. Once again, we think that this cannot be a relevant food source for urban birds. Litter bins are closed containers, which are collected every day. Thus, little or no waste is accessible to feral urban fauna by this means. However, open refuse dumps can play an important role as an infinite and predictable source of resources for several bird species, as gulls, storks, or raptors. There are only 29 large dumps in Catalonia, all of them located far from urban areas to avoid nuisance to people (for instance, only 5 within a 70km radius of Barcelona). Therefore, dumps cannot be considered a part of the urban landscape in Catalonia.

Furthermore, excepting gulls, dumps are not a suitable environment for the rest of the studied species.

Maybe the only source of human-derived food in cities worth to be mentioned is provided by retired people. Some of them have as a pastime to go to urban parks to feed bread to pigeons. Sometimes, sparrows or even parakeets may profit these situations to eat some breadcrumbs. However, this is a typical pastime in the most urbanized areas, such as Barcelona city, where feeding pigeons was also a popular attraction for tourists. However, this practice was banned some years ago in the city as part of the pigeon population control program.

In sum, as a rule, human-derived food is scarce in the urban areas of Catalonia. Even when food is available, it benefits only a few localized bird populations (e.g., pigeons in Barcelona parks). All potential sources cited before (feeders, bins, retired people) are more plausible to occur in large urban areas, as Barcelona and its metropolitan area. However, this area represents just a minor fraction of the urban areas of all Catalonia. Catalan urban areas are mainly represented by small villages and towns in the countryside. There, citizens are probably less interested on/friendly towards avian urban fauna than in big cities.

- One important point in the study remains unaddressed (it was not pointed out earlier), namely that at the moment, changes of detectability patterns, which you attribute to changes in avian behaviour, are in fact very difficult to distinguish from possible temporal shifts in human observation times. This is very loosely pointed out at the end of the discussion, but I would say it is very important to explicitly point out / discuss that changes in detectability can be caused by bias stemming from a change in the hours of work and leisure of confined observers (humans). I understand it will not be possible to disentangle this fully, but it is vital that this concern is addressed explicitly in the discussion and in the abstract.

RESPONSE: Thanks for your comment. As you say, we have been aware about this potential issue and we devoted the 6th paragraph of the discussion to discuss the pro and cons of this hypothesis. We fully agree that we cannot know what the cause is for the detectability change during the lockdown period. It could be because of a true behavioural change, an improvement in sampling conditions, or a combination of both. An improvement in sampling conditions does not seem probable, as the observers were strongly constrained in their surveys (they could watch birds only from home). In fact, the effect of sampling time showed a slower rate of discovery in all species during the lockdown year than during previous years. This provides evidence of a constrained ability of birders to find birds. As most of our volunteers have been engaged in our citizen science project since many years now, this result cannot be due to a lower profile of the observers during the lockdown period than before. Indeed, observers were the same people before and during the lockdown.

An inspection of the distribution of checklists along the day would not support the editor's hypothesis of the sampling bias. Remember that -1 is sunrise, 0 is midday, and +1 is sunset. The overall pattern is quite similar in the three groups. In all cases, most of checklists are carried out during the first half of the morning. There is another peak of observers' activity in the evening.

If we calculate the difference between the % of checklists in the lockdown and urban groups, we can make a closer examination of the editor's hypothesis:

Positive values mean more % of checklists in the lockdown group, while negative values mean more % of checklists in the historical urban group. As you can see, differences are always pretty low. Average absolute difference is just 1.26%. There are larger differences for some hourly periods (e.g., at -0.7), but such differences are never systematic. For instance, there was a slightly higher % of checklists during the lockdown at sunrise (hour = -1), but in the period 2015-19 there were more checklist just before sunrise. Thus, differences move in an erratic fashion up and down around zero. From this pattern, it can be concluded that there were no temporal changes in the observation times.

Last but not least, detectability cannot be affected by sample size. The number of observers watching birds will determine the available sample size and consequently the error of the estimated parameters (i.e. their uncertainty). For this reason, error bands were wider in the lockdown and urban group compared to the non-urban group (see fig. 2). However, sample size should not affect the parameters' magnitude, which is the truly relevant factor in this case. Detectability depends on birds' behaviour as well as on environmental conditions that affect our ability to see/hear them. In general, one can assume confidently that nor the behaviour neither the environmental conditions can be affected by the number of birdwatchers traipsing the streets.

Referee: 2

I have now read the response to the editor and reviewers as well as the new version of this ms. I do not see new or additional issues with this article than I saw in my first review and I still believe it would be an interesting paper to be published given interest that the responses of animals to lockdown has brought not only to scientists and naturalists but also to the general public.

RESPONSE: Thank you very much for reviewing again our manuscript.

However, I was in general a bit disappointed by how the authors addressed some of the comments raised in the previous review. After reading the authors response, I was left with the impression they could have put a bit more effort to address some of these issues (or at least simply acknowledge them in the text).

RESPONSE: We are very sorry to hear this, as it was not our intention to leave any previous comments and concerns properly unaddressed. We really hope that with this new version, we make all the necessary changes in the manuscript according to the reviewer suggestions and comments.

The most important issue I brought up in my first review was re. the statistical treatment of the data. I was concerned that comparing averages from data from a single year to data pooled from 5 years could result in inflated chances to detect average differences as compared with a year-by-year comparison. I realized the authors probably did that to have similar sample sizes (there were many more observations in 2020). I understand the peculiarities of the data collected for this study and this is why I did not ask the authors to repeat analyses doing year-by-year comparisons. However, I asked the authors to explicitly acknowledge this limitation, maybe in the Discussion section. Their first paragraph response to this question: *“RESPONSE: Thanks for your comment. Interestingly, when we planned the analyses, we had the opposite thought. One may think: ok, let’s compare 2020 only to 2019. However, in such pairwise comparison, it would be impossible to know whether differences are caused by conditions in 2020 or in 2019. A similar reasoning could be done for a comparison e.g. between 2020 and 2018. We pooled data from several years to have a representative baseline level or prior conditions to 2020. By pooling several years together, we avoided the potential peculiarities of each year (i.e., noise). Frankly, we think that there are more chances to get spurious differences by comparing year to year than by comparing one year against an average. In a recent paper, Derryberry et al. (2020) used pooled data from two previous years (2015-16) to make comparison with the data collected during the 2020. We firmly believe that to pool years smooths interannual variability and improves comparisons.”* I understand these arguments but remain unconvinced that the argument is statistically sound. In the second paragraph of their response to this question, they state this was in fact a sample size limitation. I believe this is the correct answer, this is fine, I just felt it was important to highlight this limitation clearly in the ms. Regarding this same question, the authors give a more humble response to my comment about Figure 1 stating that year-by-year comparisons were not possible due to limited sample sizes.

RESPONSE: We have added several sentences at the end of the 2nd paragraph of the discussion (L261-8) to explain the potential consequences: “[...] Although in urban areas the

2020 detectability pattern was patently different from the 2015-2019 baseline level, such differences could have been partially enhanced because of comparing a single year against several pooled years. Nevertheless, it would be necessary to know whether or not there is significant among years' variability in bird daily routines. If that is the case, we would expect that behavioural responses to extreme events, such as the covid-19 lockdown, would fall out of the normal variability. Unfortunately, we could not properly explore the between years' variation in detectability patterns due to sample size constrains. [...]". Furthermore, we pointed out this potential issue in the methods (L138): “[...] COVID-19 pandemics, although we could not assess variability among years. [...]"

We really appreciated the reviewer's comment about this issue, and we will consider it in the future to improve our citizen science project (e.g. convincing the volunteers to make more surveys in urban areas).

Apart from this, I believe this is a relevant, timely study that will certainly be of interest to the scientific community.

RESPONSE: Thanks a lot for your task as a reviewer, and thanks for your positive feedback on our manuscript. Despite our different opinions on certain points, we really appreciate all of your comments, as well as those from the other reviewer and the editor. They have greatly improved our study. We apologize for any misunderstandings raised from our previous reply letter.